# Transient hypoxia followed by progressive reoxygenation is required for muscle repair

Marie Quétin[1], Audrey Der Vartanian [ID][1], Christelle Dubois[1], Juliette Berthier[1], Marine Ledoux[1], Stéphanie Michineau[1], Bernadette Drayton-Libotte [ID][1], Alexandre Prola [ID][2], Athanassia Sotiropoulos[3], Frédéric Relaix [ID][1,4,5,6] & Marianne Gervais [ID][1✉]

## Abstract

**Muscle stem cells (MuSCs) are essential for skeletal muscle repair. Following injury, MuSCs reside in low oxygen environments until muscle fibers and vascularization are restablished. The dynamics of oxygen levels during the regenerative process and its impact on muscle repair has been underappreciated. We confirm that muscle repair is initiated in a low oxygen environment followed by gradual reoxygenation. Strikingly, when muscle reoxygenation is limited by keeping mice under systemic hypoxia, muscle repair is impaired and leads to the formation of hypotrophic myofibers. Sustained hypoxia decreases the ability of MuSCs to differentiate and fuse independently of HIF-1α or HIF-2α. Prolonged hypoxia specifically affects the circadian clock by increasing *Rev-erba* expression in MuSCs. Using pharmacological tools, we demonstrate that Rev-ERBα negatively regulates myogenesis by reducing late myogenic cell fusion under prolonged hypoxia. Our results underscore the critical role of progressive muscle reoxygenation after transient hypoxia in coordinating proper myogenesis through Rev-ERBα.**

**Keywords** Muscle Stem Cell; Skeletal Muscle Regeneration; Hypoxia; Circadian Clock
**Subject Categories** Musculoskeletal System; Stem Cells & Regenerative Medicine

## Introduction

Skeletal muscle exhibits robust growth (Pallafacchina et al, 2013; Moss and Leblond, 1971) and regenerative capacity (Lepper et al, 2011) which depend on the muscle stem cells (MuSCs), also known as muscle satellite cells. In adult muscle, MuSCs are maintained into quiescence and express the PAX7 transcription factor (Seale et al, 2000). Upon muscle damage, MuSCs activate and proliferate co-expressing PAX7 and MYOD before differentiating by down-regulating PAX7 and inducing MYOGENIN (also known as MYOG) to generate new myofibers (Schmidt et al, 2019). Indeed, differentiated myoblasts may either fuse to each other forming de novo myofibers or fuse with existing myotubes in the injured muscle. This additional fusion of myoblasts, called myonuclear accretion, ensures myofiber growth by increasing myonuclei number (Fukada et al, 2022). The accomplishment of this fusion process requires cell-fusion molecules, especially MYOMAKER, which is expressed in both fusogenic cells (Leikina et al, 2018). In parallel, a subset population of activated MuSCs downregulate MYOD and exit the cell cycle to self-renew the pool of quiescent PAX7+ MuSCs for future needs (Zammit et al, 2004).

Skeletal myogenesis and muscle vascularization are coupled during postnatal muscle growth (Kostallari et al, 2015) and muscle repair (Latroche et al, 2017). Vascular disruptions result in reduced oxygen ($O_2$) levels, altering cell homeostasis and contributing to many diseases, especially through hypoxia-inducible factors (HIFs) (Semenza, 2014). In skeletal muscles, HIF-1α level increases during exercise (Lindholm and Rundqvist, 2016), after systemic hypoxia (Wagatsuma et al, 2011) or during muscle injury (Yang et al, 2017) in rodents. Although the role of hypoxia, HIFs and HIF-inducible genes is well established in angiogenesis (Deveci et al, 2002; Li et al, 2011; Niemi et al, 2014), their impact on MuSC specification and myogenesis remains unclear. In vitro, most studies agree that hypoxia promotes MuSC quiescence through HIF-1α and Notch signaling (Yang et al, 2017; Gustafsson et al, 2005; Liu et al, 2012), while the impact of hypoxia and HIF-1α on myogenic cell proliferation and differentiation show conflicting results (Chaillou and Lanner, 2016; Cirillo et al, 2017; Zhang et al, 2019). In vivo, HIF-2α plays a key role in maintaining the quiescence of MuSCs in healthy muscle, as HIF-2α ablation leads to their depletion and consequent regenerative failure in mice (Xie et al, 2018). In contrast, HIF-1α KO mice in MuSCs accelerates muscle regeneration, by increasing MuSC number and their differentiation (Majmundar et al, 2015).

Molecular adaptations of skeletal muscle and satellite cells to hypoxia may also be HIF-independent. Many evidences suggest that hypoxia decreases MyoD/MyoG expressions and subsequently myogenic cell differentiation through the induction of Bhlhe40 (Wang et al, 2015) and HDAC9 (Zhang et al, 2019). Hypoxia can also alter myogenic differentiation and myotube formation by

[1]Univ Paris Est Créteil, IMRB, INSERM U955, F-94010 Créteil, France. [2]Univ Grenoble Alpes, INSERM U1055, LBFA, 38058 Grenoble, France. [3]Univ Paris Cité, Institut Cochin, INSERM, CNRS, F-75014 Paris, France. [4]École vétérinaire d'Alfort, IMRB, F-94700 Maisons-Alfort, France. [5]EFS, IMRB, F-94010 Créteil, France. [6]AP-HP, Hôpital Mondor, FHU SENEC, Service d'histologie, F-94010 Créteil, France. ✉E-mail: taurel@u-pec.fr

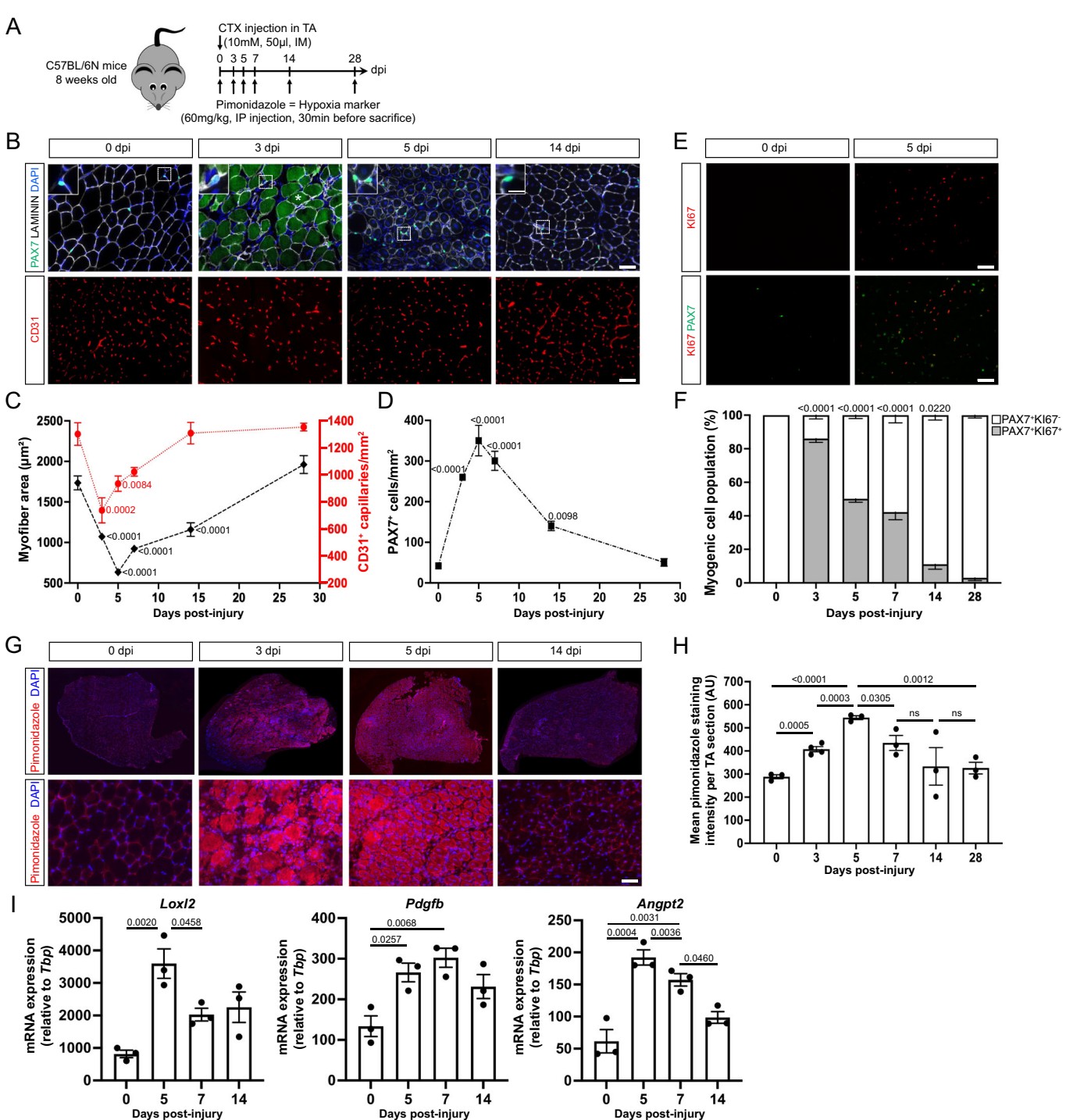

inhibiting p21 (also known as CDKN1A) independently of HIF pathway (Di Carlo et al, 2004). Moreover, it has been demonstrated that the repression of PI3K-Akt-mTOR pathway at low $O_2$ levels decreases the differentiation capacity of MuSCs (Majmundar et al, 2012) and may lead to an impairment of myofiber formation and growth (Chaillou and Lanner, 2016).

These studies reveal a wide range of factors and signaling pathways regulating myogenesis under hypoxia. However, it

remains unclear how oxygen level variation in vivo influences myogenic cell fate for effective muscle repair.

In this study, we established that skeletal muscle regeneration after cardiotoxin-induced injury is initiated in a low oxygen environment and is followed by a progressive reoxygenation. To investigate whether muscle reoxygenation following transient hypoxia is crucial for effective muscle repair, we blunted muscle reoxygenation by maintaining mice under systemic hypoxia. Our

**Figure 1.  Coordinated myo-angiogenesis during muscle repair is initiated in a hypoxic environment followed by progressive reoxygenation.**

(A) Experimental design. Acute muscle injury was induced by cardiotoxin (CTX) intramuscular (IM) injection in the tibialis anterior (TA) of 8-week-old wild-type mice. Intraperitoneal (IP) injections of pimonidazole hypoxic probe were performed during the time course of muscle regeneration at 0, 3, 5, 7, 14, and 28 days post-injury (dpi). (B) Representative pictures of PAX7 (green), LAMININ (white), CD31 (red) and nuclei (DAPI, blue) staining on CTX-injured TA cross-sections at 0, 3, 5, and 14 days post-injury. Non-specific necrotic myofiber staining (white asterisk). Scale bars: 50 μm (overviews) and 12.5 μm (insets). (C) Quantification of myofiber area (μm²) and capillary density (CD31⁺ cells/mm²) on CTX-injured TA cross-sections. (D) Quantification of PAX7⁺ cells/mm² on CTX-injured TAs cross-sections. (E) Representative pictures of PAX7 (green) and KI67 (red) staining on CTX-injured TA cross-sections. Scale bars: 50 μm. (F) Percentage of PAX7⁺ Ki67⁺ proliferative cells (%) on CTX-injured TAs cross-sections. (G) Representative pictures of pimonidazole staining (red) on CTX-injured TA cross-sections. Scale bars: 200 μm (upper panel) and 50 μm (lower panel). (H) Quantification of pimonidazole staining intensity (arbitrary unit, AU) on CTX-injured TA cross-sections. (I) Histograms showing the relative expression of HIF/HRE target genes, *Loxl2*(lysyl oxidase like 2), *Pdgfb* (platelet-derived growth factor subunit B) and *Angpt2* (angiopoietin 2), normalized to *Tbp* (TATA-box Binding Protein) on CTX-injured TA extracts. Statistics: Results are expressed as means ± SEM. For (C), (D) and (F): One-way ANOVA followed by Dunnet's post-test versus 0 dpi. $n = 3$–4 independent mice per time point. For (H) and (I): One-way ANOVA followed by Tukey's post-test. $n = 3$ independent mice per time point. Source data are available online for this figure.

results demonstrate that prolonged hypoxia impairs regenerative myogenesis and leads to long-term hypotrophic regenerating phenotype, by reducing the differentiation and fusion capacities of MuSCs. On isolated myofibers, we show that prolonged hypoxia (1%O₂) promotes the self-renewal but limits the differentiation of MuSCs, as compared to physioxia (8%O₂). In contrast, transient hypoxia followed by progressive reoxygenation (1 to 8%O₂) enhances MuSC self-renewal and restores their differentiation capacity. Using *Pax7^{CreERT2/+}*;*HIF1α^{flox/flox}* mutant mice, we established that renewal of the MuSC pool following transient hypoxia and progressive reoxygenation is HIF-1α-dependent. However, their differentiation and fusion potentials are independent of HIF-1α or HIF-2α. In contrast, transcriptomic analysis on FACS-sorted MuSCs from muscles underdoing hypoxia *versus* normoxia during regeneration, confirmed by in vitro experiments, revealed that the upregulation of Rev-Erbα under prolonged hypoxia controls the late myogenic program of MuSCs.

# Results

## Coordinated myo-angiogenesis during muscle repair is initiated in a hypoxic environment followed by progressive reoxygenation

We analyzed the spatio-temporal patterns of myogenesis, vascularization muscle oxygenation and expression of HIF target genes during skeletal muscle repair in mice that received an intramuscular cardiotoxin (CTX) injection in the *Tibialis Anterior* (TA) muscle (Fig. 1A). CTX-induced muscle injury led to disorganized skeletal muscle architecture (Fig. 1B) characterized by a drastic drop in capillar density and fiber size at 3 days and 5 days post-injury (dpi), respectively (Fig. 1B,C; Appendix Fig. S1A,B). This phenomenon was associated with a concomitant mobilization of PAX7⁺ cells to ensure muscle repair (Fig. 1B,D), as already described (Hardy et al, 2016). As expected (Latroche et al, 2017), from 5 to 28 dpi, the increase in myofiber size correlated with muscle revascularization, demonstrating synchronous myo-angiogenic coupling during muscle repair (Fig. 1B,C). Restoration of the microvascular network during muscle repair inversely correlates with PAX7⁺ cell number and their progressive return into quiescence as demonstrated by the progressive decline of proliferating PAX7⁺/KI67⁺ cells proportions from 3 to 28 dpi (Fig. 1B,D–F).

To assess whether alterations of the vascular network following muscle injury are associated with reduced oxygen levels in situ, we evaluated the kinetics of partial oxygen tension during muscle repair using a hypoxia probe, named pimonidazole (Fig. 1A). When injected in vivo, this probe is reductively activated in hypoxic cells where it forms covalent protein adducts that are revealed by immunofluorescence (Varia et al, 1998). Strikingly, the most abundant and intense pimonidazole staining is detected on CTX-injured TAs at 5 dpi, indicating that myogenic cell expansion is initiated in a hypoxic environment in situ (Fig. 1G,H). From 5 to 28 dpi, pimonidazole adduct intensity gradually declined, demonstrating a progressive reoxygenation after transient hypoxia during muscle repair (Fig. 1G,H) that correlates with progressive restoration of the perfused vascular network (Fig. 1C; Appendix Fig. S1A,B) and MuSC return into quiescence (Fig. 1B,D–F). To confirm the hypoxic status of muscle cells in vivo, we monitored the early activation of the HIF signaling pathway. We observed increased expression of known HIF-inducible target genes, such as *Lysyl oxidase homolog 2* (*Loxl-2*) (Schietke et al, 2010), *platelet derived growth factor subunit B* (*Pdgfb*) (Schito et al, 2012), and *Angiopoietin-2* (*Angpt2*) (Simon et al, 2008) transcripts, with a peak at 5 or 7 dpi (Fig. 1I).

We therefore conclude that in vivo post-injury muscle repair is associated with myo-angiogenic coupling that is initiated in a low oxygen environment followed by progressive muscle reoxygenation.

## Blocking muscle reoxygenation impairs skeletal muscle repair and leads to a hypotrophic myofiber phenotype

To address whether progressive reoxygenation after transient hypoxia is required for muscle repair in vivo, we tested the impact of systemic hypoxia exposure during skeletal muscle regeneration. TA muscles of adult mice were injured with CTX, and mice were immediately housed in a normal (21%O₂) or hypoxic (10%O₂ inhaled) normobaric atmosphere for 28 days (Fig. 2A). In vivo systemic hypoxia was validated in mice exposed to low oxygen concentration by the significant increase in their hematocrit level from 7 to 28 dpi (Appendix Fig. S2A,B). This was accompanied by sustained pimonidazole staining in regenerating TAs of hypoxic-exposed mice from 5 to 14 dpi (Appendix Fig. S2C,D), demonstrating that in vivo prolonged hypoxia slows muscle reoxygenation during repair. Interestingly, TAs exposed to prolonged hypoxia exhibited a pronounced shift towards smaller myofiber size, throughout the regenerative process (Fig. 2B,C). Morphometric

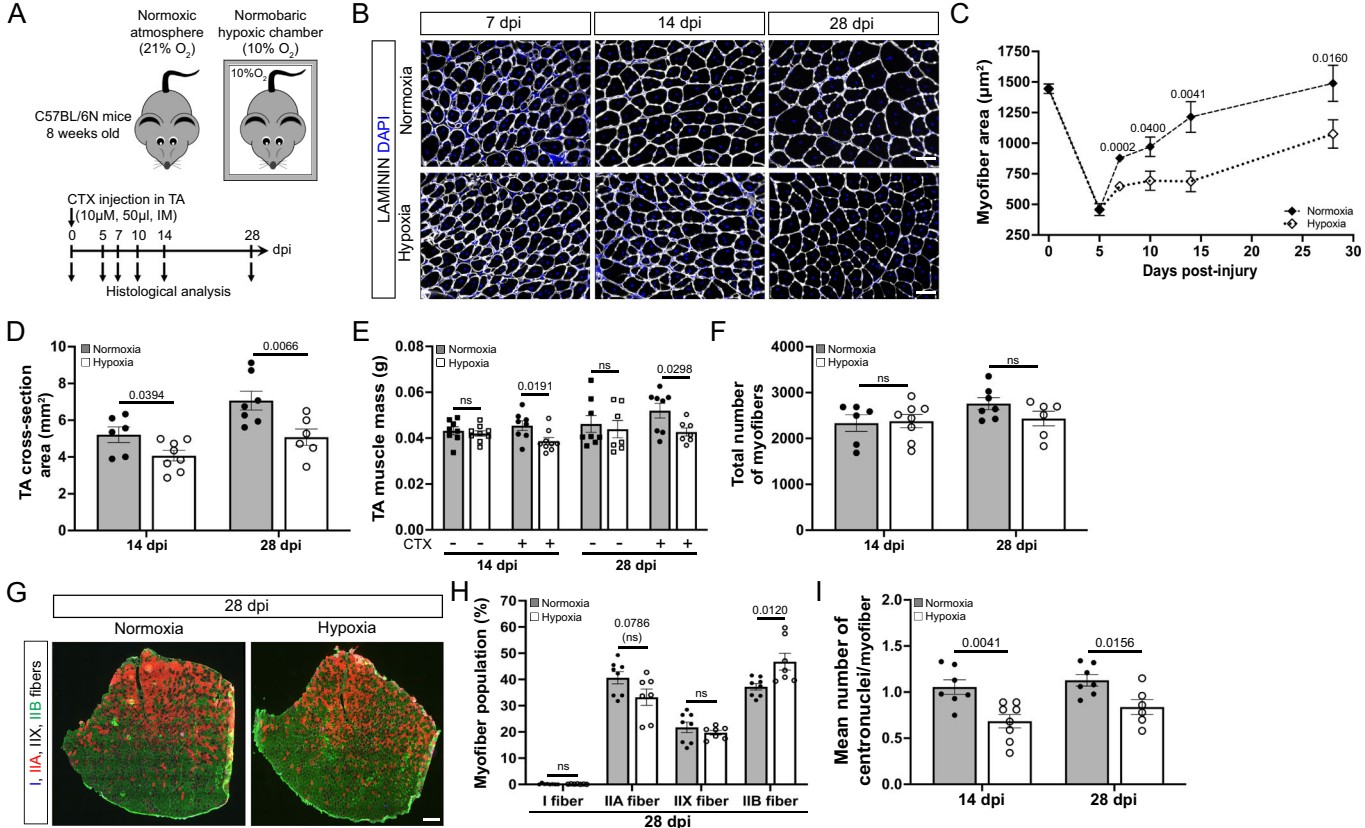

**Figure 2. Blocking reoxygenation hampers muscle repair and leads to smaller regenerated myofibers, independent of any metabolic switch.**

(A) Experimental design. Acute muscle injury was induced by cardiotoxin (CTX) intramuscular (IM) injection in the tibialis anterior (TA) of 8-week-old wild-type mice. Mice were then housed in standard normoxic atmosphere (21%$O_2$) or in a normobaric hypoxic chamber (10%$O_2$) from 0 to 28 days post-injury (dpi). (B) Representative images of LAMININ$^+$ myofiber (white) and nuclei (DAPI, blue) staining on cross-sections of CTX-injured TAs from mice exposed to normoxia or prolonged hypoxia, at 7, 14, and 28 dpi. Scale bar: 50 μm. (C) Quantification of myofiber areas (μm$^2$) on cross-sections of CTX-injured TAs from mice exposed to normoxia or prolonged hypoxia, from 0 to 28 dpi. (D) Quantification of CTX-injured TA cross-section areas of wild-type mice exposed to normoxia or prolonged hypoxia, at 14 and 28 dpi. (E) Histograms showing muscle weights (g) of CTX-injured vs contralateral non-injured TAs from mice exposed normoxia or prolonged hypoxia, at 14 and 28 dpi. (F) Quantification of the total number of myofibers in CTX-injured TAs from wild-type mice exposed to normoxia or prolonged hypoxia, at 14 and 28 dpi. (G) Representative images of type-I (blue), type-IIA (red), type-IIB (green) myofiber and laminin (white) stainings performed on CTX-injured TA transversal muscle sections of wild-type mice exposed to normoxia or prolonged hypoxia, at 28 dpi. Scale bar 200 μm. (H) Quantification of myofiber type distribution (%) on CTX-injured TAs of wild-type mice exposed to normoxia or prolonged hypoxia, at 28 dpi. (I) Quantification of the number of centronuclei per myofiber in CTX-injured TAs of wild-type mice exposed to normoxia or prolonged hypoxia, at 14 and 28 dpi. Statistics: Results are expressed as means ± SEM. For (C), (D), (E), (F), (H) and (I): Unpaired t-test. $n = 5$–8 independent mice per time point. Source data are available online for this figure.

quantification reveals a stronger difference of myofiber area at 14 and 28 dpi (Fig. 2C), at time points that mark the completion of muscle regeneration. This result correlates with significant reductions in regenerating TA cross-section area (Fig. 2D) and mass (Fig. 2E) in mice maintained under systemic hypoxia, both at 14 to 28 dpi. As there was no change in muscle mass (Fig. 2E) or myofiber area (Appendix Fig. S3B) in contralateral non-injured TAs under systemic hypoxia, the decrease in muscle mass and myofiber size in hypoxic-exposed regenerating TAs cannot be attributed to an indirect effect of systemic hypoxia, such as decreased motility or appetite in mice. Surprisingly, in hypoxia-exposed regenerating TAs, the total number of myofibers per muscle was conserved (Fig. 2F) demonstrating that the hypotrophic fiber phenotype observed under systemic hypoxia is not associated with muscular hypoplasia.

To address whether this decrease in myofiber size was not the consequence of metabolic dysregulations, we evaluated the impact of prolonged hypoxia on myofiber protein synthesis and/or fiber typing in regenerating muscles. We first compared the in vivo protein synthesis rates in whole regenerating TAs using the puromycin incorporation-based SUnSET assay, at 14 dpi (Appendix Fig. S3C), as previously described (Goodman et al, 2011). Interestingly, we observed that the protein synthesis rates were similar in regenerating TAs exposed to prolonged hypoxia compared to normoxia (Appendix Figs. S3D,E), demonstrating that blocking oxygen reoxygenation during muscle repair does not affect the hypertrophic growth of regenerating fibers. We next examined whether prolonged hypoxia affects the fiber type composition in regenerating muscles. Fiber type analysis revealed a lower proportion of type-IIA fibers and a higher proportion of

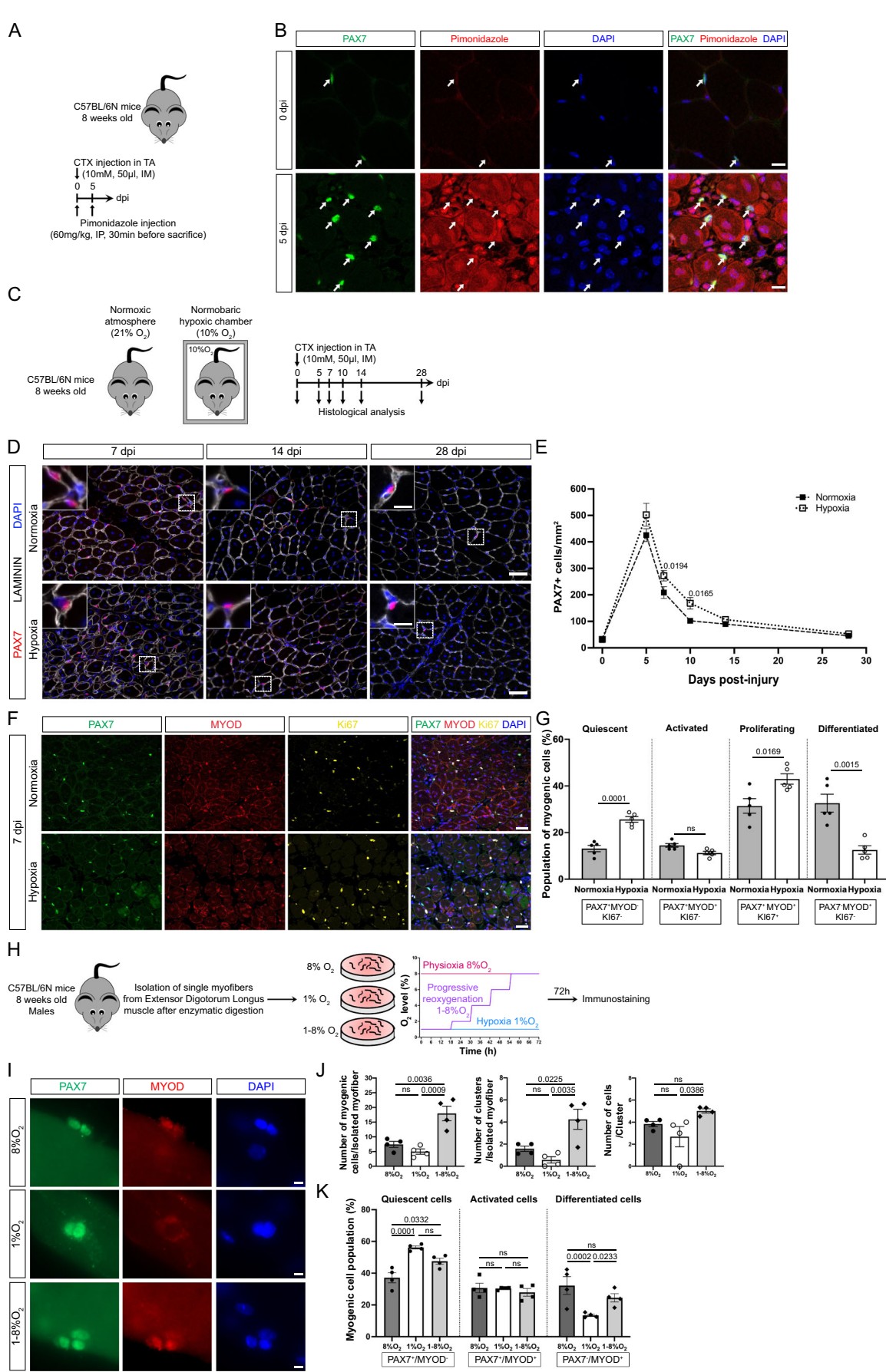

**Figure 3. The dynamics of hypoxia phases in skeletal muscle defines the myogenic fate of muscle stem cells.**

(A) Experimental design. Acute muscle injury was induced by cardiotoxin (CTX) intramuscular (IM) injection in the tibialis anterior (TA) of 8-week-old wild-type mice. Intraperitoneal (IP) injection of pimonidazole hypoxic probe were performed at 0 and 5 days post-injury (dpi). (B) Representative co-immunofluorescence staining of pimonidazole hypoxic probe (red), PAX7$^+$ cells (green) and nuclei (DAPI, blue) on CTX-injured TA cross-sections. White arrows show PAX7$^+$ cells. Scale bar: 10 μm. (C) Experimental design. Acute muscle injury was induced by CTX injection in the TA of 8-week-old wild-type mice. Mice were then housed in standard normoxic atmosphere (21%$O_2$) or in a normobaric hypoxic chamber (10%$O_2$), from 0 to 28 days post-injury (dpi). (D) Representative pictures of PAX7$^+$cells (red), LAMININ$^+$ myofibers (white) and nuclei (DAPI, blue) on CTX-injured TA cross-sections of wild-type mice exposed to normoxia or prolonged hypoxia, at 7, 14, and 28 dpi. Scale bars: 50 μm (overviews) and 12.5 μm (insets). (E) Quantification of PAX7$^+$ cells/mm$^2$ on CTX-injured TA cross-sections of wild-type mice exposed to normoxia or prolonged hypoxia. (F) Representative co-immunofluorescence staining of PAX7 (red), MYOD (red), KI67 (yellow) and nuclei (DAPI, blue) on CTX-injured TA cross-sections of wild-type mice exposed to normoxia or prolonged hypoxia, at 7 dpi. Scale bar: 50 μm. (G) Percentage of quiescent (PAX7$^+$/MYOD$^-$/KI67$^-$), activated (PAX7$^+$/MYOD$^+$/KI67$^-$), proliferating (PAX7$^+$/MYOD$^+$/KI67$^+$) and differentiated (PAX7$^-$/MYOD$^+$/KI67$^-$) cell populations on CTX-injured TA cross-sections of wild-type mice exposed to normoxia or prolonged hypoxia, at 7 dpi. (H) Experimental design. Single myofibers were isolated from Extensor Digitorum Longus (EDL) muscles of 8-week-old wild-type mice and cultured ex vivo under physioxia (8%$O_2$), prolonged hypoxia (1%$O_2$) or transient hypoxia flowed by progressive reoxygenation (1–8%$O_2$) during 72 h. (I) Representative pictures of PAX7 (green), MYOD (red) and nuclei (DAPI, blue) staining on isolated myofibers after 72 h of culture under 8%, 1% or 1–8%$O_2$. Scale bar: 20 μm. (J) Quantification of the number of cells and clusters per myofiber and the number of cells per cluster on isolated myofibers after 72 h of culture under 8%, 1% or 1–8%$O_2$. (K) Quantification of the percentage of quiescent PAX7$^+$/MYOD$^-$, activated PAX7$^+$/MYOD$^+$ and differentiated PAX7$^-$/MYOD$^+$ cells on isolated myofibers cultivated under 8%, 1% or 1–8%$O_2$ for 72 h. At 8%$O_2$, 187 cells were counted on 25 myofibers; at 1%$O_2$, 164 cells were counted on 32 myofibers; at 1–8%$O_2$, 581 cells were counted on 32 myofibers. Statistics: Results are expressed as means ± SEM. For (E): Unpaired t-test. n = 5–8 independent mice per time point and atmosphere exposure. For (G): Unpaired t-test. n = 5 independent mice per atmosphere exposure. For (J) and (K): One-way ANOVA followed by Tukey's post-test. n = 4 independent mice per $O_2$ level. Source data are available online for this figure.

type-IIB fibers in regenerating muscles exposed to hypoxia at 28 dpi (Fig. 2G,H). These data suggest that sustained hypoxia induces a transition from slow to fast newly formed myofibers following injury. However, upon systemic hypoxia, this change in myosin composition was not associated with any metabolic alterations of the myofibers, as demonstrated by similar staining for succinate dehydrogenase (SDH) activity in hypoxic- and normoxic-exposed regenerating TAs at 7 dpi (Appendix Fig. S3F). Nevertheless, given that fast myofibers are larger than slow myofibers (Liu et al, 2013), it is unlikely that the myosin switch observed under hypoxic conditions is responsible for reducing muscle mass and myofiber size. Of interest, while multi-nucleated fibers were present abundantly in normoxia-exposed mice, the number of centralized nuclei per fiber was significantly decreased, both at 14 and 28 dpi, in hypoxia-exposed regenerating TAs (Fig. 2I), that could be the consequence of an altered proliferation, differentiation and/or fusion of myogenic cells.

Collectively, our results demonstrate the essential role of reoxygenation in adult muscle regeneration. Sustained hypoxia alters myogenesis and muscle repair, possibly through an impairment of myogenic cell differentiation and/or fusion rather than inducing a metabolic adaptation of myofibers to hypoxia.

## The dynamics of oxygen levels determine the fate of MuSCs during muscle repair

To test whether systemic hypoxia impacts MuSC number and specification in vivo, we first evaluated the basal level of oxygen specifically in this cell population following CTX-injury of TA muscle. Since MuSCs reached a peak of expansion at 5 dpi (Fig. 1D), we performed co-immunostaining experiments against PAX7 and pimonidazole protein adducts at 0 and 5 dpi (Fig. 3A). At 0 dpi, the absence of pimonidazole staining suggests that quiescent PAX7$^+$ cells are not in a hypoxic state in resting muscle (Fig. 3B). In contrast, all PAX7$^+$ cells in injured muscles acquire a hypoxic state at 5 dpi, concomitantly with their expansion peak (Fig. 3B).

We thus tested the influence of sustained hypoxia on MuSC fate in vivo following CTX-mediated injury in mice housed in a normal (21%$O_2$) or hypoxic (10%$O_2$ inhaled) normobaric atmosphere for 28 days (Fig. 3C). The increase in the number of PAX7$^+$ cells at 5 dpi in CTX-injured TAs was not significantly different between mice exposed to hypoxia and control mice (Fig. 3D,E). However, under systemic hypoxia, PAX7$^+$ cells accumulated in regenerating TAs after 7 and 10 dpi (Fig. 3D,E). To unravel this phenomenon, we decided to characterize the myogenic status of hypoxic-exposed PAX7+ cells at 7 dpi using PAX7, MYOD and KI67 triple immunostaining on injured TA muscle sections. As compared to normoxia, prolonged hypoxia showed an increased number of quiescent (PAX7$^+$MYOD$^-$KI67$^-$) and proliferating (PAX7$^+$KI67$^+$) MuSCs along with a reduced number of differentiated (PAX7$^-$MYOD$^+$) myoblasts in regenerating muscles at 7 dpi (Fig. 3F,G). In contrast, systemic hypoxia did not affect the activation status of MuSCs (PAX7$^+$MYOD$^+$ cells) at 7 dpi (Fig. 3F,G). Together, these data demonstrate that suppressing reoxygenation in vivo by maintaining the mice under systemic hypoxia increases the self-renewal of MuSCs but limits their engagement towards differentiation, resulting in a transient accumulation of PAX7$^+$ cells in hypoxic-exposed regenerating muscle.

To directly address the impact of hypoxia and/or reoxygenation on the status of MuSCs, we performed complementary experiments on ex vivo floating myofibers. Myofiber culture conditions allow MuSCs to become activated, start proliferating into cluster (24–48 h), and proceed to myogenic differentiation or self-renewal of the quiescent pool at 72 h of culture (Zammit et al, 2004). Single myofiber isolated from EDL muscles were maintained for 72 h, either in physioxia (8%$O_2$), in sustained hypoxia (1%$O_2$) and in transient hypoxia followed by progressive reoxygenation (1% to 8%$O_2$) (Fig. 3H). As compared to physioxia, sustained hypoxic culture conditions had no significant impact on the overall growth of MuSCs at 72 h (Fig. 3I,J). However, prolonged hypoxia decreased the ratio of differentiating Pax7$^-$/MYOD$^+$ cells towards an increase of Pax7$^+$MYOD$^-$ self-renewing cells (Fig. 3K).

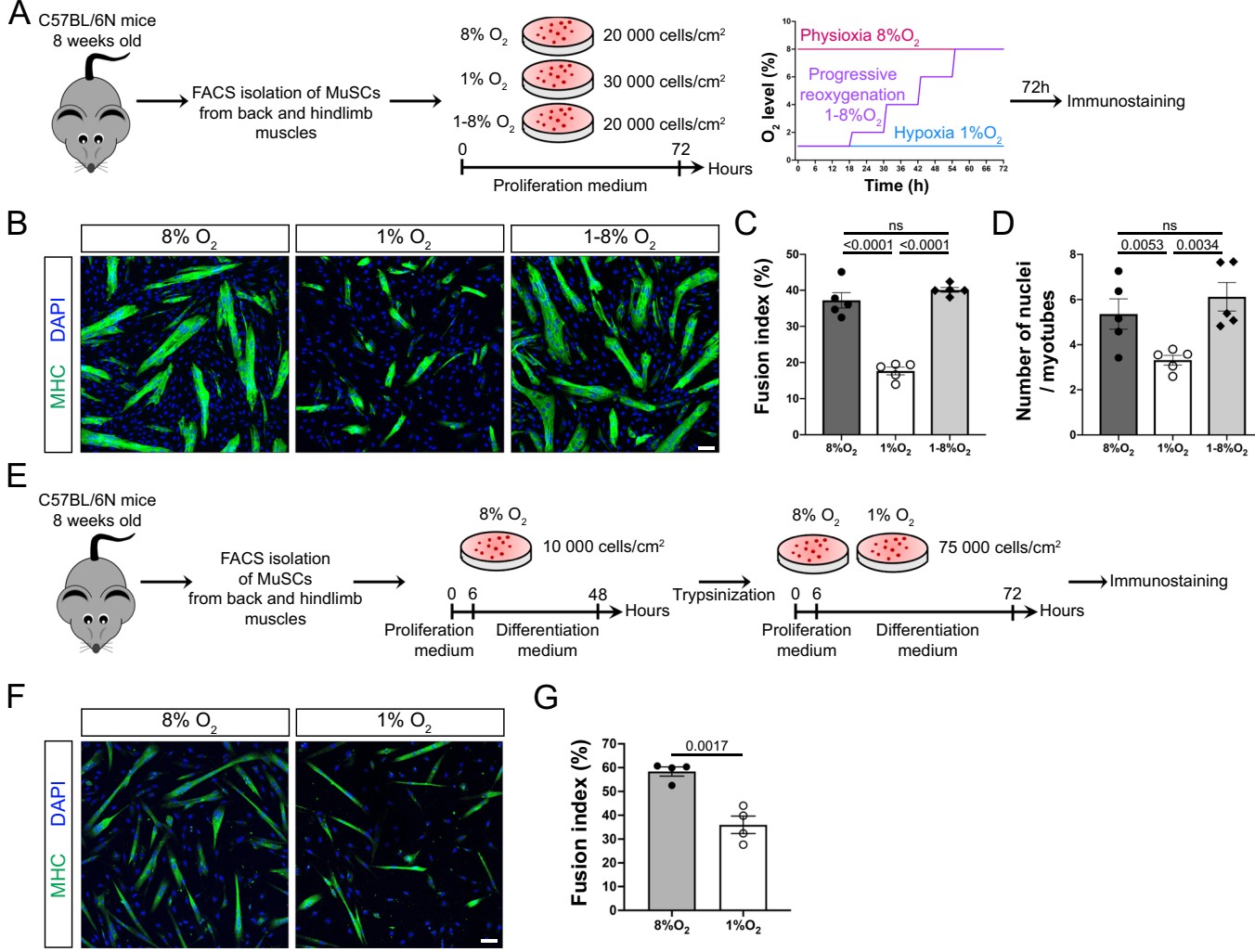

**Figure 4. Hypoxia impairs the fusion capacity of murine muscle stem cells.**

(A) Experimental design. FACS-sorted MuSCs were isolated from wild-type mice and plated with appropriate densities to ensure same confluence after 72 h of culture under physioxia (8%$O_2$), prolonged hypoxia (1%$O_2$) and transient hypoxia followed by progressive reoxygenation (1–8%$O_2$). (B) Representative pictures of MHC[+] (Myosin Heavy Chain) myotubes and nuclei (DAPI, blue) staining after 72 h of myogenic cell culture under physioxia (8%$O_2$), hypoxia (1%$O_2$) and progressive reoxygenation (1–8%$O_2$). Scale bar: 50 μm. (C) Quantification of fusion index (number of nuclei in myotubes divided by total number of nuclei, %) and of the number of nuclei per myotubes after 72 h of myogenic cell culture under 8%, 1%$O_2$ or 1–8%$O_2$. (D) Experimental design. FACS-sorted MuSCs were isolated from wild-type mice and plated at low density for 48 h under 8%$O_2$ to allow the synchronization of myoblasts, as myocytes. After trypsinization, myocytes were seeded at high density to allow their fusion after 72 h of culture under physioxia (8%$O_2$) or prolonged hypoxia (1%$O_2$). (E) Representative pictures of MHC[+] myotubes and nuclei (DAPI, blue) staining after 48 h of culture under 8%$O_2$ followed by 72 h of culture under 8% or 1%$O_2$. Scale bar: 50 μm. (F) Quantification of fusion index (%) of synchronized myoblasts after 72 h of culture under 8% or 1%$O_2$. Statistics: Results are expressed as means ± SEM. For (C) and (D): One-way ANOVA followed by Tukey's post-test. $n = 5$ independent mice per $O_2$ level. For (F): Unpaired t-test. $n = 4$ independent mice per $O_2$ level. Source data are available online for this figure.

Strikingly, transient hypoxia followed by progressive reoxygenation potentiated myogenic cell expansion and cluster density, as compared to prolonged hypoxic or physioxic culture conditions (Fig. 3I,J). Moreover, although the capacity of MuSCs to self-renew in vitro was less under transient hypoxia/reoxygenation than prolonged hypoxia, the differentiation capacity of PAX7+ cells was restored under the reoxygenation process (Fig. 3K). Both prolonged hypoxia and transient hypoxia/reoxygenation did not modulate the activation status of MuSCs, as observed in vivo (Fig. 3G). Altogether, our data suggest that blocking muscle reoxygenation through prolonged hypoxia directly disrupts the myogenic fate of

MuSCs, by enhancing their self-renewal at the expense of their differentiation.

## Hypoxia alters myoblastic fusion capacity of MuSCs independently of their commitment towards differentiation

We next addressed whether prolonged hypoxia affects myoblast cell fusion to regenerating myofibers by impairing MuSC differentiation potential or by directly impacting the fusion capacity of myogenic cells. For this, we cultured FACS-sorted GFP[+] primary

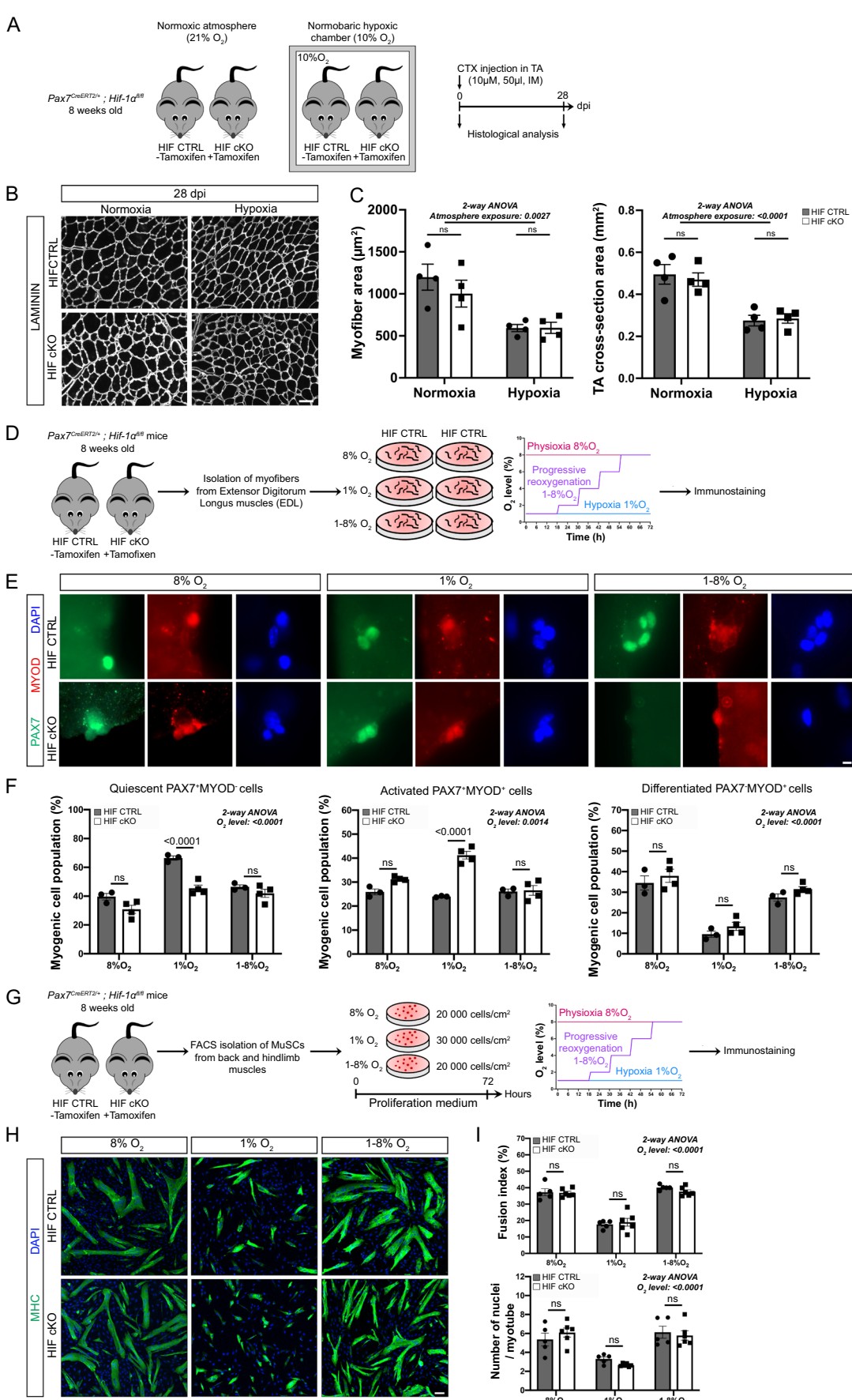

Figure panels A–I showing experimental schematics and data for HIF CTRL and HIF cKO conditions under normoxia/hypoxia and 8% O₂, 1% O₂, and 1–8% O₂ oxygen levels.

◄

**Figure 5. Impact of conditional deletion of HIF-1α in muscle stem cells on muscle repair and myogenic program.**

(A) Experimental design. Acute muscle injury was induced by cardiotoxin (CTX) intramuscular (IM) injection in tibialis anterior (TA) muscles from 8-week-old HIF CTRL (Pax7$^{CreERT2/+}$; Hif-1α$^{fl/fl}$ without tamoxifen) and HIF cKO (Pax7$^{CreERT2/+}$; Hif-1α$^{fl/fl}$ with Cre recombinase induction by tamoxifen) mice. Mice were then housed in standard normoxic atmosphere (21%$O_2$) or in a normobaric hypoxic chamber (10%$O_2$), from 0 to 28 days post-injury (dpi). (B) Representative pictures of LAMININ (white) staining on CTX-injured TA cross-sections from HIF CTRL and HIF cKO mice exposed to normoxia or prolonged hypoxia, at 28 dpi. Scale bar: 50 μm. (C) Quantification of CTX-injured TA myofiber (μm$^2$) and cross-section (mm$^2$) areas from HIF CTRL and HIF cKO mice exposed to normoxia or prolonged hypoxia, at 28 dpi. (D) Experimental design. Single myofibers were isolated from Extensor Digitorum Longus (EDL) muscles of 8-week-old HIF CTRL or HIF cKO mice and cultured ex vivo under physioxia (8% $O_2$), prolonged hypoxia (1%$O_2$) or transient hypoxia followed by progressive reoxygenation (1–8%$O_2$) during 72 h. (E) Representative staining of PAX7 (green), MYOD (red) and nuclei (DAPI, blue) on isolated myofibers from HIF CTRL or HIF cKO after 72 h of culture under 8%, 1%$O_2$ or 1–8%$O_2$. Scale bar: 20 μm. (F) Percentage of quiescent (PAX7$^+$/MYOD$^-$), activated (PAX7$^+$/MYOD$^+$) and differentiated (PAX7$^-$/MYOD$^+$) cell populations obtained on isolated myofibers from EDL of HIF CTRL or HIF cKO mice, after 72 h of culture under 8%$O_2$, 1%$O_2$ or 1–8%$O_2$. For HIF CTRL, $n = 3$ with a counting of 654 cells on 38 myofibers at 8%$O_2$, 337 cells on 43 myofibers at 1% $O_2$ and 1076 cells on 49 myofibers at 1–8%$O_2$. For HIF cKO, $n = 4$ with a counting of 643 cells on 47 myofibers at 8%$O_2$, 292 cells on 57 myofibers at 1%$O_2$ and 788 cells on 62 myofibers at 1–8%$O_2$. (G) Experience design. FACS-sorted MuSCs were isolated from HIF CTRL or HIF cKO mice and plated with appropriate densities to ensure same confluence after 72 h of culture under physioxia (8%$O_2$), prolonged hypoxia (1%$O_2$) and transient hypoxia followed by progressive reoxygenation (1–8%$O_2$). (H) Representative pictures of MHC$^+$ (Myosin Heavy Chain) myotubes and nuclei (DAPI, blue) staining after 72 h of culture under 8%, 1% or 1–8% $O_2$. Scale bar: 50 μm. (I) Evaluation of fusion index (%) and number of nuclei per myotube after 72 h of culture under 8%, 1% or 1–8% $O_2$. Statistics: Results are expressed as means ± SEM. For (C), (F) and (I): 2-way ANOVA followed by Sidak's post-tests. $n = 4$ independent mice per group and atmosphere exposure (C), $n = 3$–4 independent mice per group and $O_2$ level (F) and $n = 5$–6 independent mice per group and $O_2$ level (I). Source data are available online for this figure.

MuSCs, which spontaneously fuse in vitro in normoxic atmosphere after 72 h (Fig. 4A). Using MHC immunostaining, we showed that MuSCs maintained under prolonged hypoxia form less and smaller myotubes compared to physioxic culture conditions (Fig. 4B,C). In contrast, transient hypoxia followed by progressive reoxygenation rescued the fusion capacity of MuSCs (Fig. 4B,C).

As prolonged hypoxia reduced the differentiation status of MuSCs (Fig. 3G,K), it may be a confounding factor in vitro to access the direct impact of hypoxic conditions on myoblast fusion potential. To address this question, we seeded PAX7$^+$ cells at low density and directly cultivated them in differentiation medium, allowing the synchronization of myocytes (Fig. 4E). Cells engaged in differentiation were then seeded at high density to allow their fusion (Fig. 4E). We then quantified the fusion potential of primary myoblasts in high-density culture, in both physioxic and prolonged hypoxic conditions (Fig. 4F). Using this strategy, we conclude that prolonged hypoxia directly affects the fusion potential of MuSCs, as compared to physioxic conditions, independently of their differentiation status (Fig. 4F,G).

Since sustained hypoxia may induce metabolic changes in MuSCs and affect their myogenic fate in vivo, we conducted a non-targeted large-scale metabolomic analysis of PAX7$^+$ myogenic cells isolated from regenerated muscle at 7 dpi in mice exposed or not to prolonged hypoxia. Of the 84 metabolites identified, none were significantly altered by the hypoxic environment (Appendix Fig. S3G). This finding rules out the possibility that a shift in MuSC metabolism accounts for the observed changes in their behavior under prolonged hypoxia.

## Prolonged hypoxia impairs skeletal muscle repair by limiting the differentiation and fusion capacity of MuSCs in HIF-1α-independent manner

Overall, our results highlight the importance of transient hypoxia followed by progressive reoxygenation process in preserving the efficient myogenic differentiation and fusion abilities of MuSCs for proper skeletal muscle repair. However, the molecular mechanism by which prolonged hypoxia impairs skeletal muscle repair remains to be elucidated. In many cells, the cellular adaptation to hypoxic stress is orchestrated through the activation of the HIF signaling

pathway (Semenza, 2014). To evaluate whether blocking progressive reoxygenation by chronic hypoxia modulates the differentiation and fusion potential of MuSCs through HIF-1α signaling, we generated a MuSC-specific HIF-1α conditional knockout mouse model (HIF cKO) by combining Pax7$^{CreERT2/+}$ allele with Hif-1α$^{fl/lf}$ (Appendix Fig. S4A). We validated the efficiency of HIF-1α deletion following tamoxifen-induced Cre ablation of HIF-1α in HIF cKO MuSCs compared to HIF CTRL MuSCs at the mRNA (Appendix Fig. S4A,B) and protein levels (Appendix Fig. S4C–E). We then evaluated the specific role of HIF-1α on MuSC function in HIF cKO and CTRL mice exposed to normoxia (21% $O_2$) or systemic hypoxia (10%$O_2$ inhaled) during skeletal muscle regeneration (Fig. 5A). In normoxic atmosphere, HIF cKO mice displayed similar regenerative dynamics of myofiber growth (Fig. 5B,C), muscle size (Fig. 5C), myogenic cell expansion (Appendix Fig. S5B,D), number of centralized nuclei (Appendix Fig. S5F) and fiber type proportions (Appendix Fig. S5G,H) compared with HIF CTRL mice, at 7 or 28 dpi. Similarly, HIF cKO mice maintained in prolonged hypoxia show similar accumulation of PAX7$^+$ cells at 7 dpi (Appendix Fig. S5B,D), oxidative to glycolytic fiber-type switch (Appendix Fig. S5G,H) and muscle atrophy (Fig. 5B,C) than HIF CTRL mice at 28 dpi. These data demonstrate that HIF-1α expression in MuSCs is dispensable for muscle regeneration, regardless of the dynamics of oxygen level. To further explore the role of HIF-1α in the myogenic potential of MuSCs, we performed complementary experiments on floating myofibers isolated from HIF cKO and HIF CTRL EDL muscles (Fig. 5D). Using these experiments, we confirm that HIF-1α depletion had no impact on the differentiation state of MuSCs, whatever the oxygen level used (Fig. 5E,F). Moreover, using FACS-isolated MuSCs from HIF cKO and HIF CTRL muscles, we showed that the reduction of myoblast fusion capacity upon prolonged hypoxia was independent of HIF-1α expression (Fig. 5G–I). Of interest, we finally demonstrated that the enhancement of MuSC self-renewal under sustained hypoxia was abrogated towards increased MuSC activation when HIF-1α was depleted in MuSCs both ex vivo (Fig. 5E,F) and in vivo (Appendix Fig. S5C,E).

Recently, HIF-2α stabilization has been shown to impair MuSC differentiation and muscle repair in a mouse model of chronic obstructive pulmonary disease by pre-exposing the mice to low

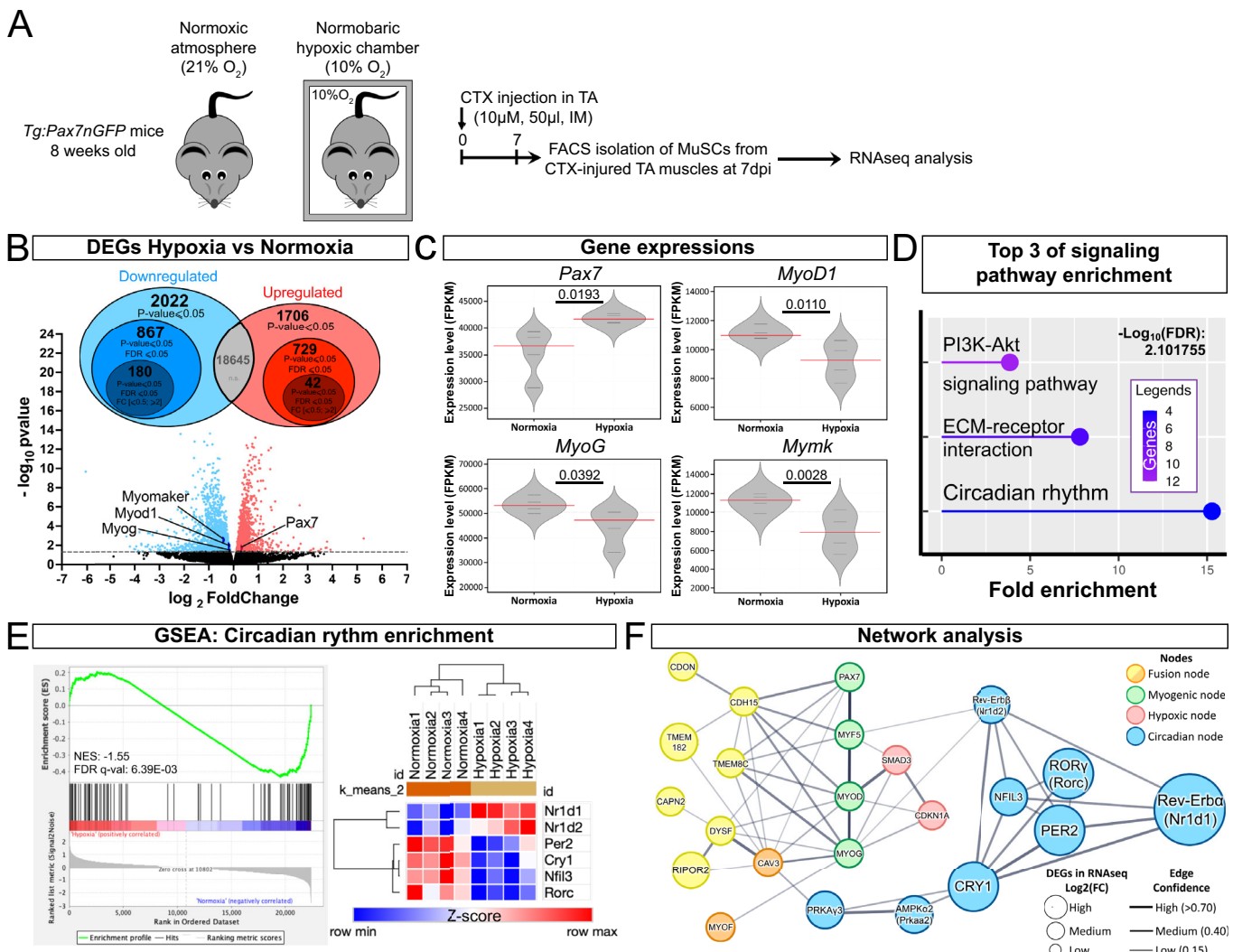

**Figure 6.** Transcriptomic analysis of PAX7⁺ cells isolated from injured *Tibialis Anterior* muscles of mice exposed to hypoxic *versus* normoxic atmosphere.

(A) Experimental design. Acute muscle injury was induced by cardiotoxin (CTX) intramuscular (IM) injection in the tibialis anterior (TA) of 8-week-old Tg:Pax7nGFP mice. Mice were then housed in standard normoxic atmosphere (21%$O_2$) or in a normobaric hypoxic chamber (10%$O_2$), from 0 to 7 days post-injury (dpi). FACS-sorted PAX7⁺ (GFP⁺) cells were isolated from CTX-injured TAs at 7 dpi to perform a bulk RNAseq analysis. $n = 4$ mice per atmosphere exposure. (B) Differential Gene Expression (DEG) of PAX7⁺cells isolated from injured TAs of mice exposed to hypoxic versus normoxic atmosphere. Volcano plot was obtained from *p*-values < 0.05. (C) Violin plots representing myogenic quiescence (*Pax7*), differentiation (*MyoD1, MyoG*) and fusion (*Mymk* = myomaker) genes significantly modulated in PAX7⁺ cells exposed to hypoxia. (D) Top 3 of signaling pathways obtained from KEGG enrichment analysis of all DEGs with FDR (False Discovery Rate) ≤ 0.05 and fold-change [0.5 ≤; ≥2]. (E) Gene set enrichment analysis (GSEA) of circadian rhythm pathway and heatmap showing the main 6 genes actors of circadian rhythm significantly (FDR ≤ 0.05 and fold-change [0.5 ≤; ≥ 2]) upregulated (red) or downregulated (blue) in our RNAseq. (F) Network representation of the protein-protein interactions of genes differentially regulated in PAX7⁺ cells exposed to hypoxia with FDR ≤ 0.05, obtained by using STRING. Statistics: Results are expressed as means ± SEM. (C) Unpaired t-test. $n = 4$ independent samples per atmosphere exposure.

oxygen content (Yin et al, 2024). To investigate whether HIF-2α could also be implicated in our phenotype, we performed in vitro experiments on FACS-isolated myogenic PAX7⁺ (GFP) cells cultured under physioxia (8%$O_2$) or hypoxia (1%$O_2$) for 72 h, in the presence or not of a HIF-2α antagonist, PT2385 (Appendix Fig. S6A). Our results showed that HIF-2α inhibition did not rescue the decreased fusion capacity of myogenic cells under prolonged hypoxia (1%$O_2$) (Appendix Fig. S6B,C).

Collectively, these results demonstrate that reducing reoxygenation by maintaining the mice under systemic hypoxia impairs skeletal muscle repair by limiting the differentiation and fusion

capacity of MuSCs independently to HIF-1α and HIF-2α, while it favors their return into quiescence through HIF-1α activation.

## REV-ERBα regulates the differentiation and fusion potential of MuSCs under prolonged hypoxia

We demonstrated that the impairment of MuSC differentiation and fusion capacities under prolonged hypoxia is not regulated by the HIF pathway. To decipher which molecular mechanisms could explain this phenotype, we performed a bulk-RNAseq on freshly isolated GFP⁺ myogenic cells from 7 dpi TAs isolated from mice

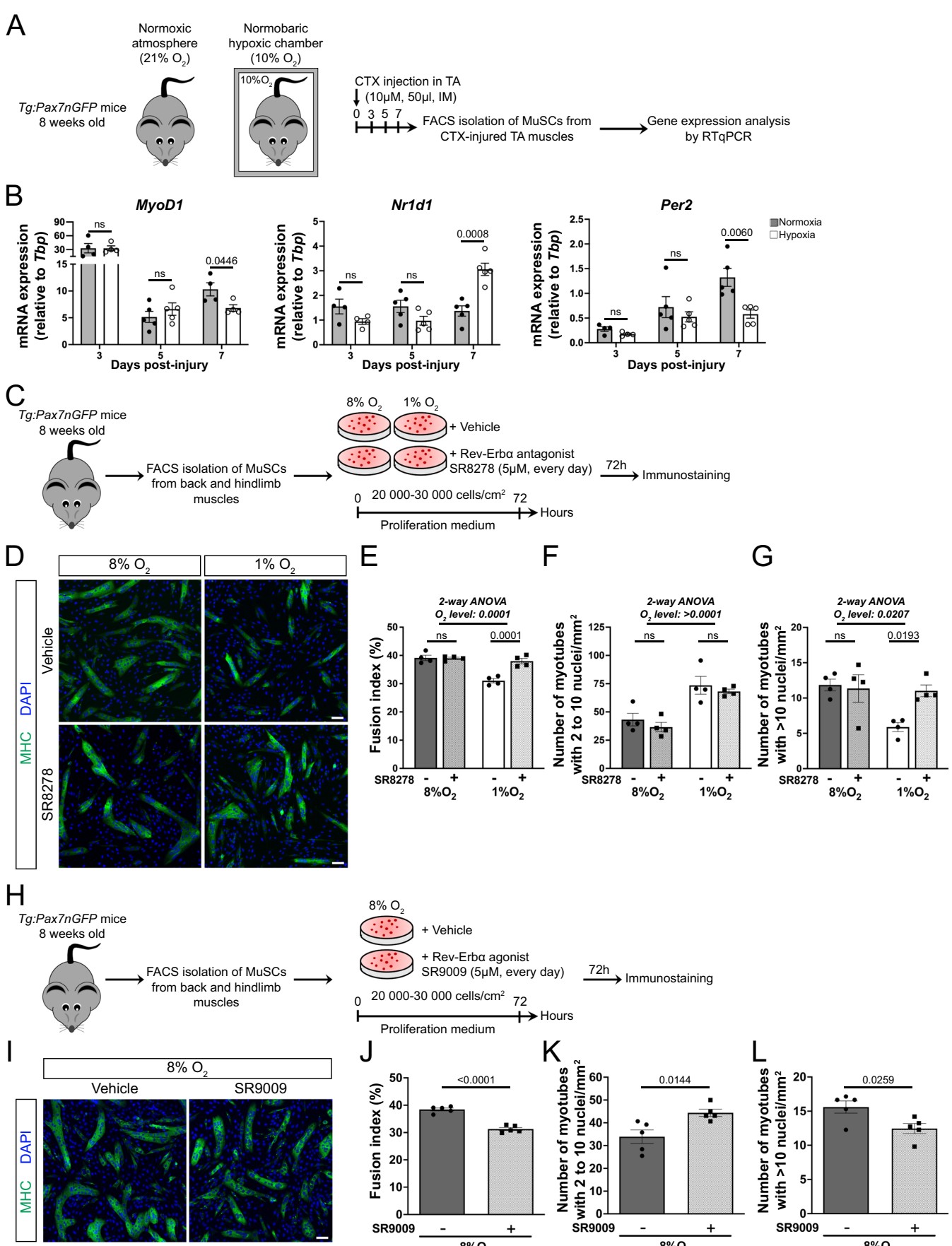

**Figure 7.  Upregulation of *Nr1d1* (Rev-ERBα) by hypoxia controls myogenic fusion.**

(A) Experimental design. Acute muscle injury was induced by cardiotoxin (CTX) intramuscular (IM) injection in the tibialis anterior (TA) of 8-week-old Tg:Pax7nGFP mice. Mice were then housed in standard normoxic atmosphere ($21\%O_2$) or in a normobaric hypoxic chamber ($10\%O_2$), from 0 to 3, 5, and 7 days post-injury (dpi). FACS-sorted PAX7+ (GFP+) cells were isolated from CTX-injured TAs at 3, 5, and 7 dpi to perform a gene expression analysis. (B) Histograms showing the relative expression of myogenic factor *MyoD1*, circadian clock effectors *Nr1d1* (nuclear receptor subfamily 1 group D member 1) and *Per2* (Period Circadian Regulator 2) normalized to *Tbp* (TATA-box Binding Protein) on FACS-sorted PAX7$^+$ (GFP$^+$) cells isolated from CTX-injured TAs at 3, 5, and 7 dpi. (C) Experimental design. FACS-sorted PAX7$^+$ (GFP$^+$) cells were isolated from 8-week-old Tg:Pax7nGFP mice and plated with appropriate densities to ensure same confluence after 72 h of culture under physioxia ($8\%O_2$) or prolonged hypoxia ($1\%O_2$), in the presence of the Rev-ERBα antagonist, SR8278 or its vehicle. (D) Representative pictures of MHC$^+$ (Myosin Heavy Chain) myotubes and nuclei (DAPI, blue) staining after 72 h of culture under 8% or $1\%O_2$, with the Rev-ERBα antagonist SR8278 or its vehicle. Scale bar: 50 μm. (E) Evaluation of fusion index (%) after 72 h of culture under 8% or $1\%O_2$, with the Rev-erbα antagonist SR8278 or its vehicle. (F) Quantification of the number of small myotubes containing between 2 and 10 nuclei/mm$^2$, after 72 h of myogenic cell culture under physioxia ($8\%O_2$) or hypoxia ($1\%O_2$), with the Rev-erbα antagonist SR8278 or its vehicle. (G) Quantification of the number of large myotubes containing more than 10 nuclei/mm$^2$, after 72 h of culture under 8% or $1\%O_2$, with the Rev-erbα antagonist SR8278 or its vehicle. (H) Experimental design. FACS-sorted PAX7$^+$ (GFP$^+$) cells were isolated from 8-week-old Tg:Pax7nGFP mice and plated with appropriate densities to ensure same confluence after 72 h of culture under physioxia ($8\%O_2$) in the presence of the Rev-ERBα agonist, SR9009 or its vehicle. (I) Representative pictures of MHC$^+$ (Myosin Heavy Chain) myotubes and nuclei (DAPI, blue) staining after 72 h of culture under $8\%O_2$, with the Rev-ERBα agonist SR9009 or its vehicle. Scale bar: 50 μm. (J) Evaluation of fusion index (%) after 72 h of culture under 8% with the Rev-erbα agonist SR9009 or its vehicle. (K) Quantification of the number of small myotubes containing between 2 and 10 nuclei/mm$^2$, after 72 h of myogenic cell culture under physioxia ($8\%O_2$) with the Rev-erbα agonist SR9009 or its vehicle. (L) Quantification of the number of large myotubes containing more than 10 nuclei/mm$^2$, after 72 h of culture under 8% with the Rev-erbα agonist SR9009 or its vehicle. Statistics: Results are expressed as means ± SEM. For (B), (J), (K) and (L): Unpaired t-test. $n = 4$–5 independent samples per condition. For (E), (F) and (G): 2-way ANOVA followed by Sidàk's post-tests. $n = 4$ independent samples per $O_2$ level and condition. Source data are available online for this figure.

exposed or not to a hypoxic atmosphere (Fig. 6A). Among the 22,373 detected genes, 3728 genes were differentially expressed (DEGs) under prolonged hypoxia, with 2022 downregulated genes and 1706 upregulated genes compared to normoxic atmosphere (p-value ≤0.05; Fig. 6B). Among those DEGs, the quiescence marker, *Pax7* was increased while *MyoD1* and *MyoG*, two myogenic differentiation factors and the fusion marker, *Myomaker* (*Mymk*) were decreased in hypoxic-exposed FACS-sorted PAX7 GFP$^+$ cells (Fig. 6C). These results confirm our in vivo and ex vivo data demonstrating that blocking reoxygenation after transient hypoxia facilitates MuSC return into quiescence but blunts their differentiation and fusion capacities.

We then selected the 222 highest modulated DEGs under prolonged hypoxia with FDR ≤ 0.05 and fold-change [0.5 ≤; ≥ 2] to perform a KEGG pathway enrichment. Among the top-3 enriched pathways was the PI3K-Akt signaling pathway (Fig. 6D), with numerous genes known to control muscle stem cell fate and repair under normoxic (Relaix and Machado, 2018; Kok and Barton, 2021) or hypoxic conditions (Gan et al, 2017; Pircher et al, 2021). Another enriched signaling pathway implicated extracellular matrix-receptor interaction genes (Fig. 6D), that are critical regulators of muscle stem cell function, skeletal muscle development and repair (Kok and Barton, 2021). Surprisingly, the most top-enriched pathway in our data set was the circadian rhythm which revealed a small but highly modulated number of DEGs (Fig. 6D). To confirm and assess which circadian rhythm genes were modulated under prolonged hypoxia, we realized a Gene Set Enrichment Analysis (GSEA) for this specific pathway (Fig. 6E). In 7 dpi injured TAs, we confirmed that MuSCs isolated from mice maintained under systemic hypoxia presented a significant modulation of genes encoding the core regulators of the circadian rhythm. Indeed, under prolonged hypoxia, circadian clock activators, such as *Cry1*, *Per2*, *Nfil3* and *Rorc* were downregulated while circadian clock repressors, such as *Nr1d1* and *Nr1d2* (coding Rev-ERBα and Rev-ERBβ, respectively) were upregulated in PAX7$^+$ cells (Fig. 6E).

To decipher the molecular interplay more deeply between the circadian rhythm and muscle repair under prolonged hypoxia, we generated a computational model of protein-protein interactions networks (PPIs) using STRING v.11 plugins (http://string-db.org). We visualized on Cytoscape software both physical and functional interaction between these DEGs for 3 clusters specifically targeting the myogenic fusion node, the core circadian clock node and the hypoxia-mediated myogenesis node. The list of genes involved in these 3 specific clusters were retrieved from GeneOntology, KEGG and Wikipathways (WP5023, 5024, 5025), respectively. According to this predictive network analysis, the edges between hypoxia and myogenesis nodes can directly explain the impact of prolonged hypoxia on the myogenic status of MuSC and its consequence on muscle repair (Fig. 6F). On the other hand, this projective network approach identifies the circadian rhythm as a key regulator of the myogenic program, either by the upstream modulation of myogenic differentiation factors or directly by affecting late myoblastic fusion (Fig. 6F). The highest regulated circadian clock DEGs is the circadian clock repressor, *Nr1d1* (Rev-ERBα) which was markedly increased by more than 2.8-fold in hypoxia-exposed PAX7$^+$cells, when muscle reoxygenation was blocked. To decipher the kinetics of the clock gene regulation upon systemic hypoxia in myogenic cells, we performed additional experiments using isolated GFP+ myogenic cells from 3, 5, and 7 dpi TAs from mice exposed or not to hypoxia (Fig. 7A). Of interest, our data showed that the upregulation of *Nr1d1* and the downregulation of *Per2* in PAX7$^+$ cells under systemic hypoxia was only evidenced at 7 dpi (Fig. 7B). At 7 dpi, the upregulation of Nr1d1 transcripts correlates with the parallel downregulation of *MyoD1* in hypoxic-exposed PAX7$^+$ cells (Fig. 7B). This result suggests that hypoxia-circadian clock interactions may reduce myogenesis through the regulation of Rev-ERBα in MuSCs.

To test our predictive model, we performed in vitro experiments on FACS-isolated MuSCs cultured under physioxia ($8\%O_2$) or hypoxia ($1\%O_2$) in the presence or not of a Rev-ERBα antagonist, SR8278 (Fig. 7C). In physioxia, inhibiting Rev-ERBα in vitro had no influence on the number or the fusion capacity of MuSCs, indicating that Rev-ERBα does not affect MuSC fate under normoxia (Fig. 7D–G). In contrast, the decreased fusion index of myogenic cells under prolonged hypoxia was fully restored when

Rev-ERBα was blocked (Fig. 7D–G). This preservation of myogenic fusion potential in the presence of Rev-ERBα inhibitor was associated with the specific increase in the number of large mature myotubes (>10 nuclei; Fig. 6G), indicating that Rev-ERBα mostly impacts the late steps of myogenic fusion process under prolonged hypoxia, notably by decreasing myonuclear accretion. We then performed the reverse experiment by incubating freshly-isolated MuSCs in the presence of a specific Rev-ERBα agonist, SR9009 under physioxia (8%O$_2$), to mimic the increase of Rev-ERBa under prolonged hypoxia (Fig. 7H). Of interest, incubation of physioxic-exposed myogenic cells with SR9009 impairs the fusion capacity of MuSCs (Fig. 7I–L) and decreases the number of large mature myotubes (>10 nuclei; Fig. 7L), reproducing the inhibitory impact of hypoxia on myogenic cell fusion. Altogether, our results demonstrate that blocking reoxygenation through sustained hypoxia impairs myogenic cell fusion through Rev-ERBα upregulation.

# Discussion

Altogether, our results demonstrate that progressive reoxygenation after transient hypoxia is essential to preserve the capacities of MuSCs to differentiate and fuse and to properly repair the damaged muscle. We further show that blocking reoxygenation by prolonged hypoxia impairs the myogenic program in a HIF-1α-independent manner but through the upregulation of Rev-ERBα.

Our work confirms that efficient skeletal muscle repair correlates with myo-angiogenesis coupling, as already shown in the literature (Latroche et al, 2017; Koike et al, 2022). However, our study is the first to decipher the dynamics of muscle oxygenation throughout the muscle repair process. We show that after acute injury, muscle repair is initiated in a low oxygen environment followed by progressive reoxygenation. In the regenerating muscle, variation of oxygen levels synchronizes with muscle revascularization. This result agrees with recent works showing that muscles are not hypoxic in the steady state but become hypoxic soon after the injury (Drouin et al, 2019). We further show that transient hypoxia following muscle injury correlates with the induction of the HIF signaling, as reported previously (Wagatsuma et al, 2011). Hence, we show that oxygen level in muscle tissue inversely correlates with the mobilization of MuSCs during muscle repair. Indeed, while in the resting muscle, quiescent MuSCs reside in a normoxic environment, proliferating PAX7$^+$ cells are in a hypoxic state after 5 dpi in the TA-injured muscle. This result is in sharp contrast with those of Xie and colleagues who claimed that quiescent MuSCs in resting muscle are intriguingly inherently hypoxic, as they expressed high level of HIF-2α at the surface of isolated myofibers (Xie et al, 2018). This affirmation is rather surprising given that quiescent MuSCs are located in the immediate vicinity of blood capillaries (Verma et al, 2018), questioning their actual hypoxic state in healthy muscle. Although it was not address in their study, cellular stabilization of HIF-2α in quiescent MuSCs may implicate hypoxia-independent mechanisms, as it has been shown in numerous other settings (Befani and Liakos, 2018). Besides, no study has yet addressed the direct impact of intramuscular oxygen levels in regulating MuSC fate and muscle repair during the regeneration process.

Of interest, we show that reoxygenation of skeletal muscle after transient tissue hypoxia is essential for proper muscle repair. Indeed, maintaining mice in a hypoxic atmosphere uncouples myo-angiogenesis affecting muscle repair, and leads to a hypotrophic muscle phenotype, mimicking muscle atrophy observed in High-landers, climbers or Chronic Obstructive Pulmonary Disease (COPD) patients exposed to long-lasting systemic hypoxia (Deldicque and Francaux, 2013). But as opposed to long-lasting hypoxia, the hypotrophic phenotype observed under prolonged hypoxia in our study is restricted to the regenerating muscle whereas the contralateral uninjured muscle remains unaffected. This result confirms that in our case, systemic hypoxia specifically impedes the muscle repair process by blocking intramuscular reoxygenation.

Whether the decrease of myofiber size could be due to a critical protein loss (Schiaffino et al, 2013), we invalidate this hypothesis as we didn't observe any change in protein synthesis rates. In the same way, this presence of smaller myofibers under prolonged hypoxia could not be explain by the glycolytic fiber-type switch from type-IIA to type-IIB, since type-IIB are the largest myofibers in mice (Liu et al, 2013). These findings, combined with the fact that hypotrophic regenerating myofibers present a reduced number of myonuclei, argue for a deregulation of MuSC myogenic abilities upon prolonged hypoxia.

In the literature, the impact of hypoxia on the fate of MuSCs has been mainly evaluated by in vitro studies and has led to controversial results (Endo et al, 2024), in particular concerning the effect of hypoxia on MuSC proliferation. Indeed, some studies showed that hypoxia promotes myogenic proliferation (Ren et al, 2010; Urbani et al, 2012) while some others demonstrate the opposite (Di Carlo et al, 2004; Launay et al, 2010; Hidalgo et al, 2014). Such a discrepancy could be attributed to the different methods used to induce hypoxia, the various degree of oxygen level tested (from 0.01% to 6%O$_2$), multiple cell types from different species and the diverse methods to quantify proliferation. Our work shows that sustained hypoxia (1%O$_2$) as compared to physioxia (8%O$_2$) tends to decrease the expansion of MuSCs ex vivo on floating myofibers. In contrast, transient hypoxia followed by progressive reoxygenation enhances myogenic cell expansion ex vivo.

Moreover, both in vivo and ex vivo, we show that prolonged hypoxia decreases the engagement of myogenic cells towards differentiation, as previously shown by others (Chaillou and Lanner, 2016; Zhang et al, 2019; Bentzinger et al, 2013). Strikingly, the differentiation defect upon sustained hypoxia is fully restored upon progressive reoxygenation ex vivo, underlying the importance of transient hypoxia and reoxygenation in properly orchestrating the myogenic program in vivo. We also establish that prolonged hypoxia (1%O$_2$) directly alters the fusion capacity of MuSCs with the formation of smaller myotubes in vitro, in agreement with previous studies (Majmundar et al, 2012; Launay et al, 2010; Hidalgo et al, 2014). While the dynamic of hypoxia during tissue repair has been overlooked, here we demonstrate that transient hypoxia followed by progressive reoxygenation rescues the fusion capacity of myogenic cells. These data suggest that blocking reoxygenation upon regeneration leads to a hypotrophic phenotype due to an impairment of both myogenic differentiation and fusion abilities.

We also establish, on single myofiber experiments, that prolonged hypoxia (1%O$_2$) promotes MuSC self-renewal, as previously reported (Liu et al, 2012; Wagatsuma et al, 2020). Moreover, transient hypoxia followed by progressive reoxygenation also favors MuSC self-renewal but to a lesser extent, underlying the crucial role of transient hypoxia for the maintenance of stemness in MuSCs.

At the molecular level, we confirm that the return into quiescence of MuSCs relies on HIF-1α stabilization, since it is lost in mice with MuSC-specific invalidation of HIF-1α, as previously demonstrated (Gustafsson et al, 2005; Yang et al, 2017). Conversely, the role of HIF signaling in modulating MuSC differentiation is still debated in literature since many in vitro studies, in line with ours, attest that hypoxia limits the commitment of MuSCs towards differentiation, independently of HIF-1α (Zhang et al, 2019; Hayot et al, 2011; Wang et al, 2015; Majmundar et al, 2012; Yun et al, 2005). Accordingly, we demonstrate that under sustained hypoxia, MuSC-specific HIF-1α-null mice exhibit the same hypotrophic regenerating myofiber phenotype than CTRL mice, indicating that prolonged hypoxia delays skeletal muscle repair independently of HIF signaling. This result is in sharp contrast with the two main studies demonstrating the essential role of HIF-1α during muscle repair (Yang et al, 2017; Majmundar et al, 2015). One explanation for this divergent result lies in the use, in their studies of two different Pax7-CreERT2 mouse lines to conditionally invalidate HIF-1α, that have both been shown to have technical limitations (Mademtzoglou et al, 2023). In our study, we used the Pax7-CreERT2 mouse line from the Kardon lab, which is the most used model to induce efficient recombination of floxed alleles in PAX7$^+$ MuSCs (Mademtzoglou et al, 2023). Moreover, our in vitro experiments show that the inhibitory effect of hypoxia on myogenic cell behavior is not mediated by HIF-2α. This contrasts with the findings of Yin and colleagues (Yin et al, 2024), who reported that HIF-2α stabilization under chronic hypoxia impairs myogenic proliferation and differentiation and muscle repair in a COPD mouse model. These discrepancies likely arise from differences in experimental design, particularly regarding the timing and the level of hypoxic exposure. While Yin and colleagues (Yin et al, 2024) applied a gradual hypoxia adaptation (from 19% to 15% atmospheric O$_2$) initiated two weeks before muscle injury, our model involved exposing mice to 10% atmospheric O$_2$ immediately after muscle injury, in order to limit muscle reoxygenation during regeneration. Moreover, Yin and colleagues model already exhibited reduced body and muscle mass, as well as smaller myofibers at the end of the hypoxia adaptation phase (Yin et al, 2024). In contrast to our design, this pre-existing muscle atrophy may have activated distinct signaling pathways under chronic hypoxia to differentially influence MuSC fate and the regenerative process.

To decipher the mechanisms independent of HIF by which prolonged hypoxia slows MuSC differentiation and fusion during muscle repair, we performed a bulk RNAseq on PAX7$^+$ cells freshly isolated from CTX-induced TAs of mice exposed to normoxia or hypoxia, at 7 dpi. Surprisingly, one of the most enriched pathways in our transcriptomic analysis is the circadian rhythm pathway. We identified a downregulation of the circadian clock activators and a parallel upregulation of the circadian clock inhibitor in PAX7$^+$ cells exposed to prolonged hypoxia. This result echoes a recent study showing that exposure to microgravity, another atmospheric environmental factor, is able to modulate MuSC fate and muscle physiology by altering the clock network, independently of light/

dark cycle (Malhan et al, 2023). In the muscle field, multiple actors of the clock machinery have been shown to regulate MuSC fate. Among these actors, the transcription factor Bmal1 (Brain and Muscle ARNT-Like 1) has been shown to regulate MYOD expression and myofiber formation to support skeletal muscle homeostasis and repair (Andrews et al, 2010). In the same line, downstream effectors of circadian rhythm, such as PER1 and PER2 seems essential for myoblast differentiation and muscle regeneration (Katoku-Kikyo et al, 2021), whether the role of CRY2 remains debated (Hao et al, 2023; Lowe et al, 2018). Indeed, while Hao and colleagues demonstrated that whole-muscle *CRY2* deletion enhance muscle repair (Hao et al, 2023), Lowe and colleagues showed that *CRY2* loss in PAX7+ cells leads to smaller myotubes and impairs muscle regeneration as a consequence of inefficient fusion process (Lowe et al, 2018). Although muscle oxygen level was not addressed in the latter study, their results echo ours. Strikingly, in our transcriptomic analysis, the highest upregulated circadian clock gene in myogenic cells maintained under prolonged hypoxia is the circadian clock negative regulator, *Nr1d1* (Rev-ERBα). Its role on myogenesis in resting or regenerating muscles remains controversial. In healthy muscle, Rev-ERBα has been shown to promote in vitro myoblast differentiation (Dadon-Freiberg et al, 2020) and positively regulate myofiber size and muscle mass in vivo (Mayeuf-Louchart et al, 2017). On the opposite, other studies suggest that Rev-ERBα inhibits myogenic differentiation in vitro by decreasing MyoD expression (Downes et al, 1995) or by inactivating Wnt pathway (Chatterjee et al, 2019). Furthermore, inhibition of Rev-ERBα has been shown to ameliorate muscular dystrophy in mice, by stimulating myoblast differentiation and enhancing myofiber formation (Welch et al, 2020; Xiong et al, 2021). Since dystrophic muscles display marked microvasculature alterations, possibly leading to a hypoxic environment (Latroche et al, 2015), the impact of Rev-ERBα on myogenesis may be dependent on intramuscular oxygen level. In favor of this argument, our data show that maintaining the mice under prolonged hypoxia during muscle repair increases Rev-ERBα in myogenic cells and that pharmacological inhibition of Rev-ERBα restores the fusion capacity of MuSCs under prolonged hypoxia. Reciprocally, incubation of myogenic cells with a Rev-ERBa agonist reproduces in vitro the inhibitory impact of hypoxia on myogenic cell fusion. Our data is the first to demonstrate a causal link between low-oxygen environment, increased Rev-ERBα and altered myogenesis.

To conclude, our work highlights the critical role of transient hypoxia followed by progressive muscle reoxygenation for proper skeletal muscle repair, involving Rev-ERBα/circadian clock pathway independently of the HIF pathway.

# Methods

**Reagents and tools table**

| Reagent/Resource | Reference or Source | Identifier or Catalog Number |
|---|---|---|
| **Experimental models** | | |
| C57BL/6N mice | Janvier Labs® | |
| Tg:Pax7-nGFP mice | Sambasivan et al, 2009 | |

| Reagent/Resource | Reference or Source | Identifier or Catalog Number |
|---|---|---|
| *Pax7*<sup>CreERT2/+</sup> mice | Murphy et al, 2011 | |
| *Hif-1α*<sup>flox/flox</sup> mice | Ryan et al, 2000 | |
| *Pax7*<sup>CreERT2/+</sup>;*Hif-1α*<sup>flox/flox</sup> mice | This study | |
| **Antibodies** | | |
| Antibodies | This study | Appendix Table S2 |
| **Oligonucleotides and other sequence-based reagents** | | |
| Genotyping primers | This study | Appendix Table S1 |
| qPCR primers | This study | Appendix Table S3 |
| Taqman probe Angpt2 | Applied Biosystems | Mm00507897_ml |
| **Chemicals, Enzymes and other reagents** | | |
| Tamoxifen | Genestil | #1324P |
| Cardiotoxin | Latoxan | #L8102 |
| Pimonidazole | Hypoxyprobe-1™ | #HP1-100Kit |
| Puromycin dihydrochloride | Sigma | #P7255 |
| Collagenase type I | Sigma | #C0130 |
| Dispase® II (neutral protease, grade II) | Roche | #04942078001 |
| Collagenase A | Roche | #11088793001 |
| Dimethyloxalylglycine | Thermo Fisher | #D3695 |
| Rev-ERBa antagonist SR8278 | MedChemExpress | #HY-14415 |
| Rev-ERBa agonist SR9009 | Sigma-Aldrich | #554726 |
| Citrate buffer pH6 | Dako | #S2369 |
| Fluoromount-G medium | Interchim | #FP-483331 |
| Lectin dylight 649 | Life Technologies Supply | #L32472 |
| RapiClear® reagent (1.52) | SunJin. Lab | #RC152001 |
| PureLink™-RNA Mini Kit™ | Invitrogen™ | #12183018A |
| RNAqueous™-Micro RNA Isolation Kit™ | Invitrogen™ | #AM1931 |
| SuperScript™ IV VILO™ Master Mix with ezDNase™ Enzyme | Invitrogen™ | #11766050 |
| SYBR™ Green Master Mix | Applied Biosystems™ | #4309155 |
| TaqMan™ Gene Expression Master Mix | Applied Biosystems™ | #4369016 |
| Pierce™ BCA Protein Assay Kit | Thermo-Fisher | #23227 |
| BoltTM LDS sample buffer 4X | Thermo-Fisher | #B0007 |
| Dithiothreitol UltraPure™ | Thermo-Fisher | #15508-013 |
| Red Ponceau-S | Merck | #1.15927 |
| SuperSignal™ West Pico Plus reagent | Thermo-Fisher | #34577 |
| **Software** | | |
| GraphPad Prism version 8.0 for MAC | La Jolla, California, USA, www.graphpad.com | |
| Zen Blue 2.0 software | Zeiss | |
| ImageJ software | https://imagej.net/ | |

| Reagent/Resource | Reference or Source | Identifier or Catalog Number |
|---|---|---|
| Imaris 9 | Oxford Instruments | |
| cSeries Capture software | Azure Biosystems | |
| Image Lab™ Software | Bio-Rad | |
| **Other** | | |
| Normobaric hypoxic chamber | Biospherix | |
| AxioImager D1 Microscope | Zeiss | |
| LSM800 confocal | Zeiss | |
| Spinning Disk microscope X1 | Zeiss | |
| Veriti® 96- Well Fast Thermal Cycler | Applied Biosystems™ | |
| StepOnePlus real-time PCR system | Applied Biosystems™ | |
| BD Influx Cell Sorter | BD Biosciences | |
| DS-11 FX spectrometer | DeNovix | |
| BoltTM 4–12% Bis-Tris Plus gel | Thermo-Fisher | #NW04122BOX |
| iBlot™ 2 PVDF membrane | Thermo-Fisher | #IB24002 |
| iBlot™ 2 Gel Transfer Device | Thermo-Fisher | #IB21001 |
| c600 imaging system | Azure Biosystems | |
| Illumina NextSeq 2000 | Illumina | |

## Mouse models

The mice used in this study were employed and maintained on C57BL/6N background: C57BL/6N (Janvier Labs®), *Tg:Pax7-nGFP* (Sambasivan et al, 2009), *Pax7*<sup>CreERT2/+</sup> (Murphy et al, 2011) and *Hif-1α*<sup>flox/flox</sup> (Ryan et al, 2000). To induce recombinaison, 5-week-old *Pax7*<sup>CreERT2/+</sup>;*Hif-1α*<sup>flox/flox</sup> mice (HIF cKO) were fed with low- phytoestrogen tamoxifen diet (Altromin 1324P, Genetsil) for 10 consecutive days. Mice fed with normal diet have been used as control (HIF CTRL). Specific forward and reverse primers used for genotyping all the mice models are listed in Appendix Table S1.

Animals were handled according to national and European community guidelines and procedures were approved by the ethics committee at the French Ministry (Project No: 16-062). Mice were exposed to common 12 h light/12 h as day–night cycle and subjected to ad libitum normal chow feeding and water. All samples were collected between 9 and 10 a.m.

## In vivo experimental design and muscle fixation

The *Tibialis Anterior* (TA) muscle was intramuscularly injected with 50 µl of cobra venom called Cardiotoxin (CTX) (10 µM, Latoxan laboratory; # L8102) on mice under isofluorane anesthesia. Then, mice have been randomly assigned and placed in a normoxic environment at 21%$O_2$ or in a normobaric hypoxic chamber (Biospherix) at 10%$O_2$ inhaled to induce a systemic hypoxia (Salyha and Oliynyk, 2023) during 3, 5, 7, 10, 14, or 28 days post-CTX injury.

In some experiments, mice have been intraperitoneally injected with a hypoxia probe named Pimonidazole (100 µl; 60 mg/kg;

Hypoxyprobe-1™; #HP1-100Kit) 30 min before sacrifice. Pimonidazole, also known as 2-nitroimidazole, forms adducts with thiol groups of proteins, peptides and amino acids of cells under hypoxic stress ($pO_2 < 10$ mmHg) that can be detected by immunostaining.

At the end of the study, adult non-injured and injured TA muscles were harvested, immediately frozen in liquid-nitrogen-cooled isopentane and sectioned transversely at 8 μm. Muscle sections were post-fixed with 4%PFA, 20 min at room temperature, washed with 1x PBS (3 times) before immunostaining as described in "Immunostaining on cells, sections and myofibers".

## SUnSET assay

SUrface SEnsing of Translation assay was performed to detect changes in protein synthesis in whole TA muscles (Goodman et al, 2011) of mice placed in a normoxic ($21\%O_2$) or hypoxic ($10\%O_2$) environment during skeletal muscle repair. Mice were intraperitoneally injected with puromycin dihydrochloride (Sigma-Aldrich, P7255) at a dose of 40 nM/g of body weight, 30 min before sacrifice. The non-injured and CTX-injured TA muscles were harvested and snap frozen in liquid nitrogen and stored at −80 °C until further use described in "Protein extraction and Western blot".

## Plasma hematocrit evaluation

Blood samples were collected through cardiac puncture from mice under isoflurane anesthesia in capillary tubes (Hirschmann®, #9100175). Tubes were centrifuged during 3 min at 10,000 rpm (Hawksley Micro-Haematocrit centrifuge) and the hematocrit levels were evaluated by measuring the proportion of red blood cells compared to total volume using an abacus.

## Isolation and culture of single myofibers

Single myofibers were isolated from EDL muscles following the previously described protocol (Zammit et al, 2004). Briefly, EDL muscles were dissected tendons to tendons and digested in a filtered solution of 0.2% collagenase (Sigma-Aldrich®; #C0130) in Dulbecco's modified Eagle's medium 4.5 g/L glucose 1x DMEM-Glutamax™-1 (Gibco™; #31966), 1% Penicillin/Streptomycin (Gibco™; #15140) for 1h30 at 37 °C. After connective tissue digestion, mechanical dissociation was performed to release individual myofibers that were then transferred to serum-coated Petri dishes for 20 min. Single myofibers were either immediately fixed in 4% PFA for 10 min before immunostaining or maintained for 72 h in myofiber growth medium containing 4.5 g/L glucose 1x DMEM-Glutamax™-1 (Gibco™; #31966), 1% Penicillin/Strepto-mycin (Gibco™; #15140), 20% fetal bovine serum (FBS; Gibco™; #10270) and 1% chicken embryo extract (CEE; MP Biomedical; CE-650-J) at 37 °C and $5\%CO_2$. For 72 h, myofibers were cultivated either at $8\%O_2$ (physioxia), $1\%O_2$ (hypoxia), or $1–8\%O_2$ (transient hypoxia followed by progressive reoxygenation: $1\%O_2$ for 18 h followed by $2\%O_2$ for 12 h, $4\%O_2$ for 12 h, $6\%O_2$ for 12 h and finally $8\%O_2$ for 18 h). Myofibers were then recovered, and fixed with 4% PFA before immunostaining for proliferation, differentiation, and quiescent markers, as described in "Immunostaining on cells, sections and myofibers".

## Muscle enzymatic dissociation

Adult hindlimb and back muscles were dissected, minced and incubated with a mix of 2.4 U/ml Dispase II (Roche, #04942078001) and 100 μg/ml collagenase A (Roche, #11088793001) in 4.5 g/L glucose 1x DMEM (Gibco™; #11995) at 37 °C on a shaking plate for 2 h. The muscle suspension was filtrated through 100-μm and 70-μm cell strainers and then spun at $50 \times g$ for 10 min at 4 °C to remove large tissue fragments. Then, the supernatant was collected, filtered through a 40-μm cell strainer and washed by centrifugation at $700 \times g$ for 10 min at 4 °C. The final pellet has been resuspended in cold 4.5 g/L glucose 1x DMEM supplemented with 0.2% bovine serum albumin (BSA, Sigma-Aldrich). The muscle cell suspension was used either to sort MuSCs by fluorescence-activated cell sorting (FACS)-sorting or directly seeded on Matrigel-coated dishes, as described in "MuSC isolation by flow cytometry".

## MuSC isolation by flow cytometry

MuSCs were sorted from dissociated muscles with the cell sorter BD Influx Cell Sorter (BD Biosciences) using either the GFP reporter for *Tg:Pax7-nGFP* mice or a cell surface labeling strategy for C57BL/6N, HIF cKO and HIF CTRL mice using anti-mouse CD45-PE-Cy7, anti-Ter119-PE-Cy7, anti-mouse CD34-BV421, anti-mouse Sca1-FITC (all from BD Biosciences, cf. Appendix Table S2) and anti-mouse integrin-α7-A700 (R&D Systems, cf. Appendix Table S2). Gating strategy for MuSC isolation using cell surface markers was as followed: Ter119$^-$, CD45$^-$, CD34$^+$, Sca1$^-$ and gating on the cell fraction, integrin-α7$^+$.

## Muscle stem cell culture

For nuclear HIF-1α translocation experiments, freshly isolated MuSCs were cultured during 72 h at $21\%O_2$ and then stimulated for 3 h at $21\%O_2$ (normoxia), $1\%O_2$ (hypoxia) or in presence with 1 mM Dimethyloxalylglycine (DMOG; Thermo Fisher; #D3695), a pharmacological mimetic of hypoxia. Cells were fixed with 4% PFA before immunostaining for HIF-1α protein, as described in "Immunostaining on cells, sections and myofibers".

For myoblastic fusion experiments, freshly isolated FACS-sorted MuSCs were plated on 8-well PCA on glass detachable plate (Sarsted, #94.6140.802) coated with Matrigel and cultured in proliferating medium containing 4.5 g/L glucose 1x DMEM-Glutamax™-1 (Gibco™; #31966), 1% HEPES 1 M (Gibco™; #15630), 20% FBS (Gibco™; #10270), 5% horse serum (Gibco™; #16050) and primocin (100 μg/ml, Invivogen; #ant-pm-05) at 37 °C and 5% $CO_2$. Cells were maintained in culture for 72 h at $8\%O_2$ (physioxia) to mimic the physioxic $O_2$ level of healthy muscle in vivo (Pircher et al, 2021; Greenbaum et al, 1997; Carreau et al, 2011), $1\%O_2$ (prolonged hypoxia) or $1–8\%O_2$ (transient hypoxia followed by progressive reoxygenation). MuSCs were seeded at 20,000 cells/cm$^2$ for 8% and $1–8\%O_2$ exposure and at 30,000 cells/cm$^2$ for $1\%O_2$ exposure. In some experiments, the HIF-2α PT2385 (10 μM; Clinisciences; #HY-12867), the Rev-Erbα antagonist SR8278 (5 μM, MedChemExpress; #HY-14415), Rev-Erbα antagonist SR9009 (5 μM, Sigma-Aldrich; #554726) or their vehicles were added every 24 h in the proliferation medium of MuSCs for 72 h. After three days of culture, cells were fixed with 4% PFA and

immunostained for cell fusion analysis, as described in "Immunostaining on cells, sections and myofibers".

In some others experiments, freshly isolated MuSCs were plated at low density 10,000 cells/cm² in 8-well PCA plate coated with Matrigel, cultured during 6 h in proliferating growth medium describe above and culture for an additional 42 h in a differentiation medium containing 4.5 g/L glucose 1x DMEM-Glutamax™-1 (Gibco™; #31966), 1% Penicillin/Streptomycin (Gibco™; #15140) and 5% horse serum (HS, Gibco™; #16050) at 37 °C, 5%CO₂ at 8% O₂, in order to favor the synchronous differentiation of myobastic cells without fusion. Differentiated cells were then trypsinized and plated at high-density (75,000 cells/cm²) in Matrigel-coated 8-well plates, maintained for 6 h in proliferation growth medium at 8%O₂ and for 72 h in differentiation medium either at 8%O₂, 1%O₂ or 1–8%O₂ as described above. This protocol was adapted from B. Chazaud's lab. Cells were then fixed with 4% PFA and immunostained for cell fusion analysis, as described in "Immunostaining on cells, sections and myofibers".

## Immunostaining on cells, sections and myofibers

Following PFA fixation, cells, muscles section or myofibers were washed three times with 1X PBS, then permeabilized and blocked at the same time with buffer containing 0.025% Tween20 for myofibers and 0.2% Triton X-100 for cells and sections. For PAX7 staining, sections were permeabilized with cold methanol, antigen retrieval was performed in boiling 10 mM citrate buffer pH 6 (Dako; #S2369) for 30 min and sections were incubated 30 min with Fab antibody. Samples were then incubated overnight at 4 °C in 2.5% BSA in 0.025% Tween20 buffer with primary antibodies listed in Appendix Table S2. Samples were washed 3 times with 0.025% PBS-Tween20 and incubated with Alexa-conjugated secondary antibodies listed in Appendix Table S2 (Life Technologies, 1/1000e) for 1 h. After washing 3 times with 1X PBS, DAPI (Sigma-Aldrich, 1/5000e) was added for 5 min at room temperature. Samples were washed 3 times with 1X PBS and slides were mounted with Fluoromount-G medium (Interchim; #FP-483331).

The staining for succinate dehydrogenase (SDH) activity was performed as previously described (Walmsley et al, 2017).

Confocal images were acquired with a Zeiss LSM800 confocal (Zeiss) or a Zeiss AxioImager D1 microscope (Zeiss) for representative pictures and analyzed with Zen Blue 2.0 software. Blinded counting was performed using ImageJ (version 1.47v; National Institutes of Health, USA, https://imagej.net).

## Tissue collection and clearing

In some experiments, mice have been retro-orbitally injected with lectin Dylight 649 (50 µl, 1 mg/ml, Life Technologies Supply; #L32472) 30 min before sacrifice.

At 0, 3, 5, and 14 dpi, TA muscles were isolated and immediately fixed in 4% PFA for 2 h at 4 °C. The tissue was washed three times for 10 min with PBS at room temperature (RT). After fixation, TA muscles were cut into 1-mm-thick slices longitudinally using a microtome (McIlwain Tissue Chopper Mct). The samples were then permeabilized with 2% Triton X-100 in PBS O/N at RT. The next day, all steps were performed in dark conditions at 37 °C in an incubator on a tube rotator at 10 rpm using 2 ml Eppendorf tubes. Tissue samples were incubated with primary antibodies (anti-

CD31, BD Biosciences; #550274) diluted in antibody dilution buffer containing 1% goat serum, 0.2% Triton X-100, and 0.2% sodium azide in PBS for 4–5 days at 37 °C, under agitation. Tissue samples were extensively rinsed with PBS for >1 h at RT, three times, under agitation. Next, the samples were incubated with a goat anti-rat secondary antibody coupled to Alexa 555 diluted in an antibody dilution buffer for 2–3 days at 37 °C under agitation in the dark. For the next 2 h, DAPI (diluted 1/5000, Sigma-Aldrich) was added and incubated in each sample. After four washes (1 h at RT), the stained samples were processed for tissue clearing by incubating them in RapiClear® reagent (1.52) (SunJin. Lab; #RC152001) for at least 12–24 h. Several z-stacks were acquired with a 25x objective on a Spinning Disk microscope X1 (Zeiss). 3D-image processing was performed using Imaris 9 (Oxford Instruments).

## Broad-scale targeted metabolomics

The metabolomic analysis was performed by the Metabolomics and Lipidomics and Lipidomics Platform of Lausanne's University, as previously described (Tsai et al, 2023). Briefly, broad-scale targeted metabolomic analysis was performed on FACS-sorted MuSCs cells isolated from CTX-injured TAs of C57BL6/N mice placed in a normoxic (21%O₂) or hypoxic (10%O₂) atmosphere. Approximately 150,000 freshly isolated MuSCs per condition were collected in cold MeOH:H₂O (9:1). Methanol extracts were dried, resuspended in 75 µl of MeOH:H₂O (4:1, v/v) and analyzed by Hydrophilic Interaction Liquid Chromatography coupled to tandem mass spectrometry (HILIC - MS/MS). Individual metabolites were processed using the Agilent Quantitative analysis software (version B.07.00, MassHunter Agilent Technologies). Relative quantification of metabolites was based on EIC (Extracted Ion Chromatogram) areas for the monitored MRM transitions or accurate masses. The obtained tables (containing peak areas of detected metabolites across all samples) were exported to "R" software http://cran.r-project.org/ and signal intensity drift correction and noise filtering (if necessary, using CV (QC features) > 30%) was done within the MRM PROBS software.

## RNA extraction and RTqPCR

Total RNA was extracted from whole muscles using PureLink™-RNA Mini Kit™ (Invitrogen™; #12183018A) or from GFP⁺ myogenic cells isolated from injured TA using RNAqueous™-Micro Kit RNA Isolation™ (Invitrogen™; #AM1931) and cDNA synthesis was performed using SuperScript™ IV VILO™ Master Mix with ezDNase™ Enzyme (Invitrogen™; #11766050), according to manufacturer's instructions. RNA quality was assessed by spectrophotometry (DeNovix DS-11 FX spectrometer). RTqPCR was performed using the Veriti® 96-well Fast Thermal Cycler (Applied Biosystems™) and real-time qPCR was performed with the StepOnePlus real-time PCR system (Applied Biosystems™) using SYBR™ Green detection tools (Applied Biosystems™, SYBR™ Green Master Mix; #4309155). Expression of each gene was normalized to *TATA Box Protein (TBP)* gene expression. Results are reported as relative gene expression ($2^{-\Delta CT}$) to *TBP*. Specific forward and reverse primers used in this study are listed in Appendix Table S3. Transcripts of Ang-2 were quantified using Taqman probe Angpt2: Mm00507897_ml (Applied Biosystems™) and TaqMan™ Gene Expression Master Mix (Applied Biosystems™, #4369016).

## Protein extraction and Western blot

Frozen TA muscles were minced on ice and lysed in RIPA buffer (Tris 50 mM, NaCl 150 mM, EDTA 1 mM, $Na_3VO_4$ 1 mM, DOC 0.5%, Triton 1%, SDS 1%, and protease inhibitor cocktail 1X, all from Sigma-Aldrich) in Lysing Matrix D tubes (MP). The samples were mixed twice during 20 s in Hybaïd Ribolyser, centrifuged at 13,000 rpm for 5 min at 4 °C. The supernatants were transferred into Eppendorf tubes, centrifuged again at 13,000 rpm for 5 min at 4 °C and collected in fresh tubes. Protein concentrations were quantified using Pierce™ BCA Protein Assay Kit (Thermo Fisher, #23227).

Lysates containing 30 µg of proteins were mixed with Bolt™ LDS sample buffer 4X (Thermo Fisher; #B0007) containing Dithiothreitol UltraPure™ (Thermo Fisher; #15508-013), boiled at 95 °C for 5 min and electrophoresed on Bolt™ 4–12% Bis-Tris Plus gel (Thermo Fisher; #NW04122BOX) at 200 V for 20 min. Proteins were then transferred onto iBlot™ 2 PVDF membrane (Thermo Fisher; #IB24002) at 20 V for 7 min using iBlot™ 2 Gel Transfer Device (Thermo Fisher; #IB21001). Membrane was stained with red Ponceau-S (Merck; #1.15927) to control the quantity of proteins deposited for each sample and then blocked for 45 min with 5% milk prepared in TBST (Tris-buffered saline supplemented with 2% Tween 20) at room temperature. Membrane was incubated overnight at 4 °C with specific mouse monoclonal primary antibody anti-puromycin (Merck; #MABE343, 1:1000) and after wash, for 1 h at room temperature with HRP-conjugated anti-mouse secondary antibody (Santa Cruz; #sc-2005, 1/10000). Between each step, the blots were washed twice for 15 min each, with TBST. Chemiluminescence was detected using SuperSignal™ West Pico Plus reagent (Thermo Fisher; #34577) and the images were acquired using Azure Biosystems c600 imaging system and cSeries Capture software (Azure Biosystems). The quantification of western-blot was performed using Image Lab™ Software (Bio-Rad).

## RNA sequencing

RNA sequencing was performed on FACS-sorted myogenic cells from dissociated TA muscles of *Tg:Pax7-nGFP* mice exposed to normoxic or hypoxic environment for 7 days after the induction of CTX injury. Libraries were constructed using an Illumina Stranded mRNA Prep (Illumina, USA) following the supplier's recommendations. Library quality and quantity were assessed using the 2100 Bioanalyzer system (Agilent Technologies) and the Qbit dsDNA HS kit (Thermo Fischer). Sequencing was performed using Illumina NextSeq 2000 platform using on a P2 cadridge for a target of 50 M reads single-ends 100 bases per sample according to the manufacturer's instruction. The RNA-seq analysis was performed with Sequana 0.11.0 (https://github.com/sequana/sequana_rnaseq). Briefly, reads were trimmed from adapters using cutadapt 3.4 then mapped to the genome assembly GRCm38 from Ensembl using STAR 2.7.3a (Dobin et al, 2013). FeatureCounts 2.0.1 (Liao et al, 2014) was used to produce the count matrix, assigning reads to features using corresponding annotation GRCm38_92 from Ensembl with strand-specificity information. Quality control statistics were summarized using MultiQC 1.10.1. Clustering of transcriptomic profiles were assessed using a Principal Component Analysis (PCA). Differential expression testing was conducted

using DESeq2 library 1.24.0 (Love et al, 2014). The normalization and dispersion estimation were performed with DESeq2 using the default parameters; statistical tests for differential expression were performed applying the independent filtering algorithm. A generalized linear model was set to test for the differential expression between conditions. Raw *p*-values were adjusted for multiple testing according to the Benjamini and Hochberg procedure and genes with an *p*-value adjusted lower than 0.05 were considered differentially expressed. Gene set enrichment analysis was performed using the Fisher statistical test for the over-representation of upregulated genes.

## Statistics

Results are presented as mean ± SEM. Each dot on graphs represents a single biological replicate. All statistical analysis and graphs were performed using GraphPad Prism® Software (version 8.0). Student test (also called T-test) was used to compare two groups. For multiple comparisons, we used one-way ANOVA followed by Dunnet or Tukey's post-test or two-way ANOVA followed by Sidak's post-test. $p < 0.05$ is considered as statistically significant.

## Data availability

The RNAseq data has been deposited in the functional genomics data collection « ArrayExpress » (BioStudies database) under the accession number: E-MTAB-16208.

The source data of this paper are collected in the following database record: biostudies:S-SCDT-10_1038-S44319-025-00679-z.

## Peer review information

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

## Acknowledgements

We would like to thank Aurélie Guguin, Adeline Henry, Odile Ruckebusch and Barbora Vrablikova for their assistance with flow cytometry. We also thank Nathalie Chevallier, Nathalie Didier, Francesca Gattazzo and Béatrice Laurent for their help with the hypoxic incubators as well as Serge Adnot and Shariq Abid for the use of the normobaric hypoxic chamber. We thank Camille Laisne, Diana Gelperowic and Damien Fois for animal care and animal facilities: EP3 (IMRB) and CDTA (TAAM, CNRS – UPS44). The RNAseq analysis was performed by E. Turc and E. Kornobis of Biomics Platform, C2RT, Institut Pasteur, Paris, France. The metabolomic analysis of MuSCs was performed by H. Gallart Ayala of Metabolomics and Lipidomics and Lipidomics Platform, Faculty of Biology and Medicine, University of Lausanne, Switzerland. This work was supported by fundings from Association Française contre les Myopathies (AFM) via TRANSLAMUSCLE II (PROJECT 19507) and from Agence Nationale de la Recherche (ANR) via MyoStemVasc (ANR-17-CE14-0018-01).

## Author contributions

**Marie Quétin**: Conceptualization; Data curation; Formal analysis; Methodology; Writing—original draft; Writing—review and editing. **Audrey Der Vartanian**: Data curation; Formal analysis; Methodology; Writing—original draft. **Christelle Dubois**: Data curation; Formal analysis; Methodology. **Juliette Berthier**: Data curation; Methodology. **Marine Ledoux**: Data curation; Formal analysis; Methodology. **Stéphanie Michineau**: Data curation; Methodology; Writing—original draft. **Bernadette Drayton-Libotte**: Methodology. **Alexandre Prola**: Data curation; Formal analysis; Methodology. **Athanassia Sotiropoulos**: Writing—original draft; Writing—review and editing. **Frédéric Relaix**: Funding acquisition; Writing—original draft; Writing—review and editing. **Marianne Gervais**: Conceptualization; Data curation; Formal analysis; Supervision; Funding acquisition; Methodology; Writing—original draft; Project administration; Writing—review and editing.

Source data underlying figure panels in this paper may have individual authorship assigned. Where available, figure panel/source data authorship is listed in the following database record: biostudies:S-SCDT-10_1038-S44319-025-00679-z.

## Disclosure and competing interests statement

The authors declare no competing interests.

