## [Peer Review File · EMBO Reports]

Transient hypoxia followed by progressive reoxygenation is required for muscle repair

Marie QUETIN, Audrey DER VARTANIAN, Christelle DUBOIS, Juliette BERTHIER, Marine LEDOUX, Stephanie MICHINEAU, Bernadette DRAYTON-LIBOTTE, Alexandre PROLA, Athanassia Sotiropoulos, Frédéric Relaix, and Marianne GERVAIS

Corresponding author(s): Marianne GERVAIS (taurel@u-pec.fr)

Review Timeline:	Submission Date:	12th Aug 25
	Editorial Decision:	22nd Sep 25
	Revision Received:	18th Nov 25
	Accepted:	8th Dec 25

Editor: Esther Schnapp

**Transaction Report: This manuscript was transferred to
EMBO reports following peer review at Review Commons.**

**Review
COMMONS**

Review #1

1. Evidence, reproducibility and clarity:

Evidence, reproducibility and clarity (Required)

SUMMARY

Quéting et al investigated the dynamics of oxygen levels during the skeletal muscle regeneration following sterile damage and its impact on muscle repair. They combined in vivo and ex-vivo model systems, together with genetic and pharmacological manipulations. They found results consistent with the fact that a dynamic oxygenation process, hypoxia during the early phase followed by reoxygenation, are involved in muscle repair. Prolonged hypoxia leads to defective myogenesis and muscle repair. These activities appear to be mediated by modulation of Rev-ERB α levels. Collectively, the study provides intriguing insight regarding the role of oxygen in muscle repair.

MAJOR COMMENTS

1. In Figure 1, the 5 days post CTX injury is too late to claim that "myogenic cell expansion is initiated in a hypoxic environment". Indeed, at day 5 myofibers are already regenerated, although immature. To support their claim, the authors should perform analyses and quantification of Pax7+, Pax7+Ki67+ and hypoxia at earlier timepoints.
2. In Figure 2B, a larger number of mononuclear cells is present in hypoxia mice. Is hypoxia affecting the number/activity of extra-muscular cells important for muscle regeneration like for example FAPs, macrophages, etc?
3. In Figure 5H, the myotubes formed by HIF-1 α cKO appear thinner than control myotubes. Is myotube size affected by lack of HIF1 α ?
4. The choice of the 7 days post CTX for the RNA-seq is odd. Indeed, at that timepoint there are obvious histological abnormalities in hypoxia mice. Hence, it is highly likely that many DEGs are simply secondary to the defect in regeneration and not directly linked to hypoxia exposure. This is probably the reason why the authors found so many (close to 4K) DEGs. To focus on the genes closely-associated to the primary defect, the authors should have performed the RNA-seq at an earlier timepoint, in which minimal histological defects were present. While repeating the RNA-seq would be costly and time consuming, the authors could at least address this issue by RT-qPCR. Are muscle stem cell fate, repair, and

circadian clock genes significantly altered 3 and 5 days after CTX injury in hypoxia vs normoxia?

5. Given that compounds have frequently off-target effects, the authors must independently support their Rev-ERBa findings by performing genetic manipulations, at least ex-vivo.

6. A recent study (PMID: 38333911), which was not cited by the authors, reports muscle atrophy and weakness, impaired muscle regeneration, and increased fibrosis in hypoxia exposed mice. Intriguingly, this was due to impaired MuSC proliferation and differentiation following HIF-2 α stabilization under hypoxia. Hence, the authors should investigate if HIF-2 α plays any role in the phenotypes they describe. For example, is HIF-2 α a regulator of circadian clock genes expression?

****Referees cross-commenting****

The other reviewers raised very relevant issues and I fully agree with their comments. In particular, I concur with Reviewer #3 that in several instances the evidence provided by the authors does not support the conclusions made.

2. Significance:

Significance (Required)

SIGNIFICANCE

There is a limited knowledge regarding the role of oxygen supply during tissue differentiation and repair. In the muscle field, there are conflicting reports in the literature. This study combines genetic, pharmacological and oxygen manipulations both in vivo and ex-vivo to investigate the role of oxygen during regeneration following sterile skeletal muscle injury. The results are very intriguing and potentially relevant both for muscle, but possibly also for other tissue repair.

Aspects of the study that must be improved concern the role of HIF-1 α and HIF-2 α in the process, and the characterization of the molecular mechanism through which Rev-ERBa is regulated by oxygen and regulates muscle repair.

- AUDIENCE: specialized, basic research, translational research; results could potentially extend beyond the muscle field.

- FIELD OF EXPERTISE: muscle differentiation, muscular dystrophy, gene expression regulation.

3. How much time do you estimate the authors will need to complete the suggested revisions:

Estimated time to Complete Revisions (Required)

(Decision Recommendation)

Between 1 and 3 months

4. Review Commons values the work of reviewers and encourages them to get credit for their work. Select 'Yes' below to register your reviewing activity at Web of Science Reviewer Recognition Service (formerly Publons); note that the content of your review will not be visible on Web of Science.

Yes

Review #2

1. Evidence, reproducibility and clarity:

Evidence, reproducibility and clarity (Required)

The manuscript, Transient hypoxia followed by progressive reoxygenation is required for efficient skeletal muscle repair through Rev-ERBa modulation, revisits the role of hypoxia in skeletal muscle regeneration after acute injury. They first nicely demonstrate, using the pimonidazole hypoxia probe, that during regeneration skeletal muscle is transiently hypoxic at 5 days post injury (DPI). Then they show skeletal muscle regeneration is impaired in mice housed in a hypoxic (10% O₂) chamber; the regenerated muscle mass is smaller, due to smaller regenerated myofibers and there is a shift in myofiber type so that there are more IIB myofibers. In addition, at 7 DPI when mice are raised in a hypoxic environment there is a shift in muscle stem cells so that they are more proliferative and fewer have differentiated. Ex vivo experiments culturing muscle stem cells in association with EDL myofibers in 1% O₂, as compared with 8% O₂, also led to fewer differentiated Pax7-MyoD⁺ cells, but could be restored if O₂ was subsequently increased to 8%. They also found that low oxygen inhibited myoblast fusion in vitro. They then tested, via Pax7CreERT2^{+/+};HIF-1afl/fl, whether

HIF-1a signaling mediated the response of muscle stem cells to hypoxia in vivo. Surprisingly, they found that loss of HIF-1 α did not impair myofiber regeneration in normoxic or hypoxic conditions, but they do provide some data suggesting that HIF-1a is required for the hypoxic-induced increase in Pax7+MyoD- muscle stem cells. Bulk RNA-seq analysis of 7 DPI muscle from mice housed in normoxic versus hypoxic conditions uncovered the interesting mis-regulation of circadian rhythm associated genes - in particular, the circadian clock repressor Rev-ERBa. Using a pharmacological antagonist of Rev-ERBa they show in culture that blocking Rev-ERBa (in contrast to loss of HIF-1a) rescues the fusion defect of muscle stem cells cultured in 1% O₂. Conversely, they show that a Rev-ERBa agonist inhibits fusion in 8% O₂. Altogether, the paper provides interesting new data on the controversial role of hypoxia and HIF-1a as well as data suggesting a connection between hypoxia and circadian rhythm genes. The data is logical and well presented, and the paper will be of strong interest to the regeneration and skeletal muscle research communities. I have two major comments and a list of smaller suggestions to improve the manuscript.

Major comments:

1. In vivo experiments (presented in Figures 2, 3, 5, 6, 7) house mice in hypoxic (10% oxygen) chambers, and the authors suggest that this blocks the progressive reoxygenation of skeletal muscle during regeneration. Surprisingly, the authors do not test when the mice are in hypoxic chambers whether, in fact, skeletal muscle is hypoxic at homeostasis and whether during regeneration muscle experiences prolonged hypoxia. The obvious experiment would be to use the pimonidazole probe on skeletal muscle sections of muscle at homeostasis and at 0, 5, 6, 14, and 28 DPI CTX injury in mice housed in hypoxic chambers. Without some demonstration that skeletal muscle oxygenation is changed when the mice are housed in hypoxic chambers, it is impossible to interpret these experiments.

2. The authors claim that reducing reoxygenation by maintaining the mice under systemic hypoxia impairs skeletal muscle repair by limiting the differentiation and fusion capacity of MuSCs in HIF-1a-independent manner, while it favors their return into quiescence through HIF-1a activation. They provide some in vitro evidence that Hif1a is required for the high levels Pax7+MyoD- muscle stem cells in 1% O₂. They should also show that the elevated levels of Pax7+ muscle stem cells at 7 DPI (seen in Fig. 3D-G) requires HIF1a via analysis of Pax7CreERT2/+;HIF-1a^{fl/fl} mice.

Minor comments:

1. Please provide a reference for the pimonidazole probe. Reference 26, Hardy et al., is not the right one.
2. Please provide references that *Loxl-2*, *Pdgfb*, and *Ang2* are HIF-inducible target genes.
3. Fig. 2C shows changes in average myofiber diameter. How was this calculated? Is this the largest diameter? Is there a reason that cross-sectional area was not measured (the more standard measurement)? Also, generally this type of data is shown as bar graphs - which is how these data are shown in Fig. 5C. Please also show the data in Fig. 2C as bar graphs.
4. Please provide reference for 8% O₂ being physioxia in culture.
5. Fig.5 should also quantify the number of centronuclei/myofiber (as in Fig. 2I) for Pax7CreERT2/+;HIF-1^{af/fl} mice 14 and 28 DPI - to further demonstrate that differentiation defects in hypoxia are HIF-1a independent.
6. Please provide a graphical model of your research findings.
7. There are many typos and verb tense issues. Please fix these. The most amusing is Stinkingly in the Discussion.

****Referees cross-commenting****

I think several important issues are raised by myself and reviewer 3. First, the authors need to explain and support their use of 10% O₂ hypoxia in vivo chambers and 1% O₂ for hypoxic in vitro experiments. Second, the authors have not demonstrated that reoxygenation of muscle is prevented in mice raised in hypoxic chamber. There are questions about how well the pimonidazole probe is working (the widespread expression at 5 dpi in Fig. 1E suggests there may be specificity issues) and this probe is also not shown for muscle from mice living in hypoxic chambers. Another method of demonstrating hypoxia in muscle tissue would be useful.

2. Significance:

Significance (Required)

The paper provides interesting new data on the controversial role of hypoxia and HIF-1a as well as data suggesting a connection between hypoxia and circadian rhythm genes.

This paper will be of interest to researchers studying the role of hypoxia on regeneration and also to researchers studying muscle regeneration.

3. How much time do you estimate the authors will need to complete the suggested revisions:

Estimated time to Complete Revisions (Required)

(Decision Recommendation)

Between 3 and 6 months

Yes

Review #3

1. Evidence, reproducibility and clarity:

Evidence, reproducibility and clarity (Required)

The manuscript by Quéting et al "Transient hypoxia followed by progressive reoxygenation is required for efficient skeletal muscle repair through Rev-ERBa modulation" describes the nature of muscle stem cell (MuSC) differentiation within its hypoxic niche using in vivo, ex vivo and in vitro methodologies. Approaches to limit oxygen in a regenerating model of muscle injury showed that muscle oxygenation is necessary for proper muscle repair. They found that the lack of oxygen is associated with the formation of hypotrophic myofibers, due to the inability of MuSCs to differentiate and fuse. Their findings show that the phenotype was independent of HIF-1a. However, RNA-seq of MuSCs 7 day post injury from prolonged hypoxia was shown to have significantly increased circadian clock gene Rev-erba expression. Pharmacological inhibition of Rev-erba during hypoxia rescued the

myogenic phenotype. Contrarily, the use of Rev-erba agonist in normoxia impaired the fusion capacity of MuSCs and decreases the number of large mature myofibres. This manuscript is well written and very easy to follow. Though, there are certain shortcomings outlined below. Sometimes the evidence provided does not support the conclusions made. For example, more rigour should be performed to state that there is a self-renewal phenotype.

****Major issues****

1. In Figure 1, why were these timepoints chosen? Is the hypoxia more severe between days 0 and 5 (i.e. when MuSCs begin their activation).

2. "From 5 to 28 dpi, pimonidazole adduct intensity gradually declined, demonstrating a progressive reoxygenation after transient hypoxia during muscle repair (Fig. 1E and 1F) that correlates with progressive restoration of the vascular network (Fig. 1C) and MuSC return into quiescence (Fig. 1B and 1D)." For this statement, correlating these events to MuSC returning to quiescence might not be appropriate. As Figure 1D shows all the Pax7+ cells, it does not reflect whether they are quiescent. Thus, the timelines might not actually match up with the proportion of self-renewed MuSCs?

3. The manuscript cites far too many review articles (at least half) and not primary sources. Also, some citations are misrepresented. For example: Reference #13 does not show that HIF-1alpha level increases during muscle injury in rodents, Reference #15 shows fusion is impaired in hypoxic c2c12 cells, not promotion of quiescence, Reference #22 does not support the claim that hypoxia induces myostatin expression, only that myostatin inhibits MyoD expression.

4. Figure 1E and 1F, does the dye intensity change with it being more accessible to the muscle during early injury as opposed to later recovery. Also, when using the probe for hypoxia determination, the whole tissue is fluorescing intensely suggesting potential non specificity. It would be prudent to use markers of hypoxia on western blots or gene expression to corroborate this data.

5. a) It is well known that CTX injury does not cause damage to the vasculature but directly to the muscle (Tatsumi et al doi:10.1002/stem.2639; Ramadasan-Nair et al doi:10.1074/jbc.M113.493270; Ohtsubo et al. doi: 10.1016/j.biocel.2017.02.005; Wang et al doi: 10.3390/ijms232113380). How do the authors reconcile their findings that there is vasculature damage with CTX (Fig. 1C).

b) Moreover, the endothelial cell staining (Fig. 1B) appears to be unchanged in the time course of injury. To prove vascular damage this data should be corroborated, for example with lectin perfusion.

6. Problems with Figure 3J. There are data points with zero clusters/isolated myofibres suggesting that the hypoxic environment caused MuSCs to not activate from quiescence. There are several outliers for example at 1% there is a zero reading that makes the data significant.
7. In Figure 1G, Loxl2 after 14 days appears to be significant, as the error bars at 0 and 14 days do not overlap and thus it does not return to normal. An n=3 is not sufficient, as one of the data points at 14 days appears to be an outlier (the data stretching from 1500 to 3000).
8. In Fig. 2C and 2D, there are no control CSA and myofiber diameter experiments for keeping the mice in hypoxia over 14 and 28 days without injury.
9. For Figure 3K, how can self-renewing MuSCs be distinguished from MuSCs that never activated? Especially in the 1% O₂ condition where few clusters formed. How does hypoxia influence activation? A 4hr or 8hr timepoint is necessary, as well as 24hrs. Also, for Figure 5E and 5F, it is possible that HIFcKO allowed the cells to activate normally, thus explaining the shift from quiescence to activation in the read-outs. This further highlights the importance of analyzing earlier timepoints. One cannot state that these cells are self-renewing or returning to quiescence without performing experiments on earlier timepoints.
10. The data for Figure 4 does not suggest that transient reoxygenation is required "for proper skeletal muscle repair" as stated by the authors only that reoxygenation has rescued the phenotype in the primary myoblasts. There is no hypoxia in the control (8% O₂) for regeneration to occur (Fig. 2B).
11. One cannot rule out metabolic dysregulation. It's true that glycolytic fibers are generally larger than oxidative, it is likely that that alone does not explain the difference in fiber size. However, the fact that the fibers are more glycolytic does suggest a metabolic shift in the muscle (which was the aim of the experiment), which could also shift MuSC character altering their behaviour. How are MuSCs metabolically responding to hypoxia?
12. In Figure 2, how can one be sure that reoxygenation is blocked by the hypoxic chamber? Reduced O₂ levels will induce hypoxia, but one cannot state that it blocks reoxygenation without further validation such as using pimonidazole as in Fig. 1E. If reoxygenation is blocked, then pimonidazole staining should remain consistent throughout the injury.
13. For Figure 3G, is a sum appropriate for the graph? Proportions would be more appropriate as cell number is not equal as shown in figure 3E. Can Pax7+/MyoD+ be defined as differentiated? By day 7, many MuSCs will have fused and be expressing MyoG, which is not accounted for by these definitions. Did systemic hypoxia increase self-renewal or impair activation? How can you distinguish these two?
14. In Figure 6A, while it is interesting that Pax7 levels are elevated in hypoxia and differentiation and fusion markers are down at 7days, it does not necessarily mean that self-renewal is increased. It might suggest that the hypoxic cells might have never activated or might have differentiated precociously. Are any cell cycle genes down regulated? Any

other genes involved in quiescence altered?

15. The use of pimonidazole in Fig. 1E shows the staining within fibers (many with centrally located nuclei). These nuclei are differentiating and not representative of expanding MuSCs. How do the authors reconcile these MuSCs as part of their population.

****Minor Problems****

1. In the introduction, the line "Vascular alterations result in reduced oxygen (O₂) levels, disrupting cell homeostasis and contributing to many diseases" is not always true as vascular alterations do not always result in reduced oxygen levels. For example, in angiogenesis there is no reduction of O₂. This line should better reflect this.

2. In the introduction, Paragraph 2, line 9 change "quiescence through HIF-1 α " to "quiescence through HIF-1 α ".

3. Paragraph 3, line 8: "lead" instead of "leads"

4. It is not sure how important the connection between capillary density and Pax7+ cell number is. Both are presumed to occur at the same time in muscle, so both will recover concurrently. To state that it is a coupled response is overstating the evidence presented.

5. Figure 1B the colour-labels for Pax7 and Dapi overlap with the border.

6. In the Introduction, the following sentence does not follow the previous sentence: "In vivo, Majmundar and colleagues show that HIF-1 α in MuSCs negatively regulates myogenesis by decreasing myogenic differentiation".

7. In the Introduction, the following statement is not accurate "Hypoxia can also alter myogenic differentiation and myotube formation by inhibiting p21 (as known as p21 and CDKN1A) that leads to an accumulation of the retinoblastoma protein Rb24", for what was found in the reference. The authors should correct this statement.

8. Paragraph 3, line 5: "as known as p21 and CDKN1A" should perhaps read "also known as CDKN1A"

9. The following statement is not supported by the results: "Strikingly, the most abundant and intense pimonidazole staining is detected on CTX-injured TAs at 5 dpi, indicating that myogenic cell expansion is initiated in a hypoxic environment in situ (Fig. 1D-1F)." MuSCs are activated and expanding from time zero to 5 days according to Figure 1D.

10. "...Since glycolytic fibers are larger than oxidative fibers, ..." citation missing

11. An inconsistent finding is that the authors show that protein synthesis rates are normal between normoxia and hypoxia of regenerating muscle (suppl. Fig. 1E), yet the capacity of protein synthesis is found to be higher in oxidative muscle fibres compared to glycolytic fibers (Van Wessel et al, doi: 10.1007/s00421-010-1545-0), which are formed during regeneration (Fig. 2G and 2H).

12. Some figure legends that describe graphs do not denote the number of samples or

mice used.

13. In Figure 1C, 1D and 1F what is being compared to obtain statistical significance?

14. The font size of many figures is too small to follow.

15. Confusion for the results of figure 3G. Labels in the text do not reflect the labels in figure (which cannot be read anyway because the font is too small). Why is Ki67 used as a marker for activation versus proliferation.

16. The physiological O₂ concentration is 8%, do the authors know what the hypoxic O₂ concentration is in the injured environment. Why did they choose hypoxic O₂ concentration at 1% for ex vivo and invitro experiments? Why did they choose 10% for the in vivo experiment?

17. For Figure 2H it is not appropriate to state that type IIA ratio was reduced with hypoxia, as the results show no statistical significance.

18. For Figure legend 3K, are the cell number/fiber the sums per one mouse or the sum from all mice combined for each condition?

19. For Figure 3B and 3E "concomitantly with their proliferation peak" seems to imply that hypoxia in Pax7+ cells peaks alongside proliferation, but the evidence doesn't support that conclusion. More timepoints would be needed to show that 5 dpi is truly the peak of hypoxia in Pax7+ cells.

20. For Figure legend 4E, should read "MHC" not "MCH"

21. In Figure 4C there is no gap between the significance bar.

22. In Figure legend 5G, "Experience design" should read "Experimental design"

23. Representative images Fig 3I and 5E are poor quality.

24. Confusing statement "In the same way, this presence of smaller myofibers under prolonged hypoxia could not be explain by the glycolytic fiber-type switch from type-IIA to type-IIB, as observed in pathological context of COPD or peripheral arterial disease (PAD), since type-IIB are the largest myofibers in mice."

****Referees cross-commenting****

I agree with the thoughtful reviews and issues raised by Reviewers 1 and 2. I do not have anything more to add.

2. Significance:

Significance (Required)

General Assessment: This manuscript is well written and easy to follow. It rigorously investigates the influence of oxygenation on MuSC behaviour. The authors utilize in vivo, ex vivo, and in vitro models to support their study and executed their work to a high degree. A

limitation is that all experiments are only performed in mice and might not be applicable in humans. In addition, some claims made by the authors were over-reaching. The study can be improved by further validating some of the authors' claims, as has been suggested in the review.

***Advance:** This study is the first to report the effect of hypoxia on MuSCs in an ex vivo culture and in vivo injury model using a hypoxia chamber. This study helps clarify the role of HIF-1 α on MuSC behaviour by suggesting that it does have a role in MuSC fate decisions. Finally, the authors make a novel link between circadian rhythm and MuSC behaviour in hypoxia.

***Audience:** A specialized audience that is interested in myogenesis, muscle stem cells, and/or hypoxia will be interested in this study. It highlights the important role of oxygen in muscle regeneration and may help researchers understand the role of oxygen in MuSC fate decisions.

3. How much time do you estimate the authors will need to complete the suggested revisions:

Estimated time to Complete Revisions (Required)

(Decision Recommendation)

Between 3 and 6 months

Yes

Full Revision

Manuscript number: RC- 2024-02511

Corresponding author(s): Dr. Marianne GERVAIS

1. General Statements [optional]

We would like to thank the editor and reviewers for their positive and constructive comments on the manuscript. Our response to the various points raised by the reviewers is provided below. We have incorporated these comments into our updated manuscript, significantly improving our article in the process. Our study is the first to demonstrate a causal link between low-oxygen environment, Rev-ERB α /circadian clock pathway and myogenesis.

Our answers, indicated in blue, will be detailed point by point.

Reviewer #1

Major comments

1. In Figure 1, the 5 days post CTX injury is too late to claim that "myogenic cell expansion is initiated in a hypoxic environment". Indeed, at day 5 myofibers are already regenerated, although immature. To support their claim, the authors should perform analyses and quantification of Pax7+, Pax7+Ki67+ and hypoxia at earlier timepoints.

→ We thank the reviewer for raising this point. As requested, the analysis of muscle oxygenation has been performed at an earlier time point, i.e., 3dpi and the quantification of PAX7⁺ and PAX7⁺/Ki67⁺ cells has been performed at all timepoints. The results are presented in Figure 1 and detailed in the results section (page 5, lines 13-16). Our data show that the accumulation of PAX7⁺ cells peaks at 5dpi (Figure 1B-1D), with a maximum proliferation rate at 3dpi (Figure 1E-1F). Furthermore, our data show that the intensity of the pimonidazole adduct already increases at 3dpi compared to baseline, with a maximal peak at 5dpi. Taken together, these data confirm that the myogenic cell expansion is initiated in a hypoxic environment with a maximal hypoxic intensity at 5dpi.

2. In Figure 2B, a larger number of mononuclear cells is present in hypoxia mice. Is hypoxia affecting the number/activity of extra-muscular cells important for muscle regeneration like for example FAPs, macrophages, etc?

→ In Figure 2B, the images in the former manuscript were not representative of the regenerative muscle phenotype of hypoxic mice, which show clear evidence of muscle mononuclear cell accumulation. To avoid confusion, the representative images of hypoxic muscles in Figure 2B have been replaced.

Regarding the effect of prolonged hypoxia on extra-muscular cells, such as macrophages or FAPS, although interesting, we believe that this question is beyond the scope of the manuscript. Nevertheless, to partially address the reviewer's concern, we quantified the number of CD68+ macrophages in hypoxic vs normoxic TAs at 7 and 14dpi (Figure #1). Our results show that prolonged hypoxia led to an accumulation of CD68+ cells in regenerating TAs, which is consistent with a delay in the resolution of inflammation and muscle repair in hypoxic mice. For more clarity, we have decided not to include these new results in the revised manuscript.

Figure #1: Percentage of CD68+ cell area on CTX-injured TA cross-sections of wild-type mice exposed to normoxia or prolonged hypoxia, at 7 and 14 dpi. Data are means±SEM.

3. In Figure 5H, the myotubes formed by HIF-1α cKO appear thinner than control myotubes. Is myotube size affected by lack of HIF1 α?

→ To address the reviewer remark, we compared the size of HIF-1α cKO vs control myotubes. We show that HIF-1α depletion doesn't affect the size of myotubes after 72h, whatever the level of oxygen under which the myogenic cells are cultured (Figure #2).

Figure #2: Evaluation of myotubes area (μm²) after 72h of culture under 8%, 1% or 1-8% O₂. Data are means±SEM.

4. The choice of the 7 days post CTX for the RNA-seq is odd. Indeed, at that timepoint there are obvious histological abnormalities in hypoxia mice. Hence, it is highly likely that many DEGs are simply secondary to the defect in regeneration and not directly linked to hypoxia exposure. This is probably the reason why the authors found so many (close to 4K) DEGs. To focus on the genes closely-associated to the primary defect, the authors should have performed the RNA-seq at an earlier timepoint, in which minimal histological defects were present. While repeating the RNA-seq would be costly and time consuming, the authors could at least address this issue by RT-qPCR. Are muscle stem cell fate, repair, and circadian clock genes significantly altered 3 and 5 days after CTX injury in hypoxia vs normoxia?

→ We chose to perform our analysis at 7dpi, as this was the only timepoint at which we observed both a significant accumulation of PAX7+ cells under prolonged hypoxia (Figure 3E) and a concomitant impairment in their differentiation capacity (Figure 3G). Nevertheless, to address the reviewer's concern, we examined the kinetics of muscle stem fate and circadian clock gene expression by RT-QPCR at earlier time points than 7 dpi. As reported in Figure 7B, our data showed that *MyoD*, *Nr1d1* and *Per2* mRNA levels were altered in hypoxic regenerating TAs but only at 7 dpi; no change in their expression level was observed at 3dpi and 5 dpi. Overall, these data support our choice of 7dpi for RNA-seq analysis. These new data have been included in the revised manuscript (Figure 7B and results section page 12, lines 19-24).

5. Given that compounds have frequently off-target effects, the authors must independently support their Rev-ERB α findings by performing genetic manipulations, at least ex-vivo.

→ We agree with the reviewer that REV-ERB antagonist could have off-targets, but our *in vitro* data are rather convincing as REV-ERB antagonists and agonists had the exact opposite effects on the modulation of muscle stem cell fate under prolonged hypoxia. We deliberately chose this pharmacological approach as a suitable alternative to the use of REV-ERB α mutant mice, which are not available to us, and because it aligns with the 3Rs rule.

6. A recent study (PMID: 38333911), which was not cited by the authors, reports muscle atrophy and weakness, impaired muscle regeneration, and increased fibrosis in hypoxia exposed mice. Intriguingly, this was due to impaired MuSC proliferation and differentiation following HIF-2 α stabilization under hypoxia. Hence, the authors should investigate if HIF-2 α plays any role in the phenotypes they describe. For example, is HIF-2 α a regulator of circadian clock genes expression?

→ We thank the reviewer for bringing to our attention this study which was published during the process of our paper submission (Yin et al., 2024 ; PMID: 38333911) that explores the role of HIF-2 α in mediating muscle atrophy and impaired muscle regeneration in a model of chronic obstructive disease obtained by pre-exposing the mice to prolonged hypoxia. Although HIF-2 α can form a heterodimer with BMAL1 to modulate the circadian clock (Mello et al., 2024 ; PMID: 38895384), the role of HIF-2 α as a direct regulator of the expression of circadian clock genes is not known. To investigate the potential involvement of HIF-2 α in our phenotype, we performed additional experiments using freshly isolated mouse satellite cells cultured in hypoxic conditions, in the presence or absence of an HIF-2 α antagonist, PT2385 (used at the same dose as in the aforementioned study). Our results show that HIF-2 α inhibition did not rescue the impaired fusion capacities of myogenic cells under prolonged hypoxia (Supplemental Figure 6A-C). Our results thus demonstrate that altered muscle stem cell fate through prolonged hypoxia is independent of

HIF-2 α signaling. These new data have been included in the revised version of our manuscript (Supplemental Figure 6A-C ; Summary section page 2, line 9 ; Introduction section page 3, lines 23-26 ; Results section page 10, lines 26-33 and page 11, lines 1-4 ; Discussion section page 16, lines 16-29 ; Methods section page 36, lines 3-4).

Reviewer #2

Major comments:

1. In vivo experiments (presented in Figures 2, 3, 5, 6, 7) house mice in hypoxic (10% oxygen) chambers, and the authors suggest that this blocks the progressive reoxygenation of skeletal muscle during regeneration. Surprisingly, the authors do not test when the mice are in hypoxic chambers whether, in fact, skeletal muscle is hypoxic at homeostasis and whether during regeneration muscle experiences prolonged hypoxia. The obvious experiment would be to use the pimonidazole probe on skeletal muscle sections of muscle at homeostasis and at 0, 5, 6, 14, and 28 DPI CTX injury in mice housed in hypoxic chambers. Without some demonstration that skeletal muscle oxygenation is changed when the mice are housed in hypoxic chambers, it is impossible to interpret these experiments.

→ We thank the reviewer for raising this important point. To address the reviewer's concern, we examined the kinetics of skeletal muscle oxygenation during regeneration in normoxic vs hypoxic environments using the pimonidazole hypoxic probe. Our data show that when mice were housed in hypoxia, the intensity of the hypoxic probe staining remained greater at all time points in regenerating TAs (Supplemental Figure 2C-D), demonstrating that muscle reoxygenation during regeneration is hampered when the mice are housed in hypoxic chambers. These new data have been incorporated in the revised manuscript (Supplemental Figure 2C-D ; results section page 6, lines 9-12).

2. The authors claim that reducing reoxygenation by maintaining the mice under systemic hypoxia impairs skeletal muscle repair by limiting the differentiation and fusion capacity of MuSCs in HIF-1 α -independent manner, while it favors their return into quiescence through HIF-1 α activation. They provide some in vitro evidence that Hif1 α is required for the high levels Pax7⁺MyoD⁻ muscle stem cells in 1% O₂. They should also show that the elevated levels of Pax7⁺ muscle stem cells at 7 DPI (seen in Fig. 3D-G) requires HIF1 α via analysis of Pax7CreERT2^{+/+};HIF-1 α fl/fl mice.

→ As requested by the reviewer, we have quantified the percentage of total PAX7⁺ and quiescent PAX7⁺KI67⁻ cells 7dpi in HIF CTRL and HIF cKO mice under normoxic or hypoxic environment (Supplemental Figure 5C). Our data show that although the percentage of PAX7⁺ cells remains unchanged in HIF cKO mice compared to HIF CTRL mice regardless of the oxygen environment, the percentage of quiescent PAX7⁺KI67⁻ satellite cells is reduced in HIF cKO mice exposed to prolonged hypoxia. This confirms that the return of satellite cells to quiescence under prolonged hypoxia requires HIF-1 α *in vivo*. These new data have been included in the revised manuscript (Supplemental Figure 5C ; results section page 10, lines 22-25).

Minor comments:

1. Please provide a reference for the pimonidazole probe. Reference 26, Hardy et al., is not the right one.

→ We apologize for this mistake; the correct reference is “Varia et al., 1998” (PMID: 9826471). Manuscript reference has been changed in the revised version (References section, page 43, lines 13-15).

2. Please provide references that Loxl-2, Pdgfb, and Ang2 are HIF-inducible target genes.

→ New references demonstrating that Loxl-2 (Schietke et al., 2010 ; PMID: 20026874), Pdgfb (Schito et al., 2012 ; PMID: 23012449) and Ang2 (Simon et al., 2008 ; PMID: 18720385) are HIF-inducible targets genes have been included in our manuscript (Results sections, page 5, lines 28-31 ; References section, page 43, lines 16-24).

3. Fig. 2C shows changes in average myofiber diameter. How was this calculated? Is this the largest diameter? Is there a reason that cross-sectional area was not measured (the more standard measurement)? Also, generally this type of data is shown as bar graphs - which is how these data are shown in Fig. 5C. Please also show the data in Fig. 2C as bar graphs.

→ In the initial submission of our manuscript, myofiber diameters are estimated and calculated from corresponding myofiber area. For more clarity, in the revised manuscript, we have replaced the calculated myofiber diameter by the myofiber area, expressed in μm^2 (Figure 1C, 2C, 5C).

Regarding the presentation of our data in Figure 2C, we believe that showing our result as a bar graph alters the readability and interpretation of our results on the effect of hypoxia on myofiber regeneration kinetics.

4. Please provide reference for 8% O₂ being physioxia in culture.

→ In the literature, most *in vitro* studies have compared the impact of cells cultured in hypoxia (1-2% O₂) vs. ambient atmosphere (21% O₂). However, with the exception of lung tissue, it is well established that *in vivo* tissue O₂ tensions are significantly lower than ambient O₂ levels, although they vary depending on the tissue and species. For example, in healthy skeletal muscle, the physioxic O₂ content is known to vary between 4 and 10% *in vivo* in mammals (Greenbaum et al., 1997; PMID: 9127263; Carreau et al., 2011; PMID: 21251211). This is why we chose an O₂ concentration of 8% for our *in vitro* and *ex vivo* cultures to mimic the physioxic oxygen level found in healthy muscle *in vivo*. We have added a sentence to the Material and Methods section (page 35, line 34 and page 36, line 1) to justify our choice of 8% oxygen as the physioxic reference *in vitro*.

5. Fig.5 should also quantify the number of centronuclei/myofiber (as in Fig. 2I) for Pax7CreERT2/+;HIF-1 α fl/fl mice 14 and 28 DPI - to further demonstrate that differentiation defects in hypoxia are HIF-1 α independent.

→ As requested, we have quantified the number of centronuclei/myofiber for HIF cKO and HIF CTRL mice exposed to normoxia and hypoxia at 28dpi (Supplemental figure 5F). Our data show that regenerated TAs from HIF CTRL mice exposed to hypoxia exhibited less centronuclei than mice exposed to normoxia. This decreased number of myofiber centronuclei under prolonged hypoxia was not reversed in HIF cKO mice,

demonstrating that differentiation defects under hypoxia are HIF-1 α -independent. These new data have been included in the revised manuscript (Supplemental figure 5F ; Results section page 10, lines 9-10).

6. Please provide a graphical model of your research findings.

→ Please find below a proposal of graphical abstract for our research findings.

7. There are many typos and verb tense issues. Please fix these. The most amusing is Stinkingly in the Discussion.

→ We apologize for the numerous mistakes in the typos and verb tense in the former manuscript. These issues have been corrected in the revised manuscript.

Reviewer #3

Major issues

1) In Figure 1, why were these timepoints chosen? Is the hypoxia more severe between days 0 and 5 (i.e. when MuSCs begin their activation).

→ As requested by the reviewer, we have added an earlier timepoint, 3 dpi for the hypoxic probe staining. Our data show that a continuous increase in pimonidazole adduct intensity between day 0 and 5 dpi, with a maximal peak intensity at 5dpi (updated Figure 1G-H). These data confirm that the level of hypoxia is

maximal in regenerating TAs 5 days post-injury. This new timepoint has been included in the revised manuscript (Figure 1G-H).

2) "From 5 to 28 dpi, pimonidazole adduct intensity gradually declined, demonstrating a progressive reoxygenation after transient hypoxia during muscle repair (Fig. 1E and 1F) that correlates with progressive restoration of the vascular network (Fig. 1C) and MuSC return into quiescence (Fig. 1B and 1D)." For this statement, correlating these events to MuSC returning to quiescence might not be appropriate. As Figure 1D shows all the Pax7+ cells, it does not reflect whether they are quiescent. Thus, the timelines might not actually match up with the proportion of self-renewed MuSCs?

→ To address the reviewer's request, we have performed the quantification of quiescent PAX7+/Ki67- vs proliferative PAX7+/Ki67+ myogenic cells at all time points. We also included an earlier time point at 3dpi. Our data show that the proliferation level of myogenic cells peaks at at 3dpi (~90% of PAX7+ cells also being Ki67+), as previously described (Cutler et al., 2022 ; PMID: 35733848). In contrast, quiescent satellite cells displayed the opposite kinetics, representing only ~10% of PAX7+ cells at 3 dpi, followed by a continuous increase up to 28 dpi. These new data further support our conclusion that muscle regeneration is initiated in a hypoxic environment, followed by progressive reoxygenation, which correlates with both revascularization of the tissue and the return of MuSCs to quiescence. These findings have been incorporated into the revised manuscript (Figure 1E-F ; results section page 5, lines 13-16).

3) The manuscript cites far too many review articles (at least half) and not primary sources. Also, some citations are misrepresented. For example: Reference #13 does not show that HIF-1alpha level increases during muscle injury in rodents, Reference #15 shows fusion is impaired in hypoxic c2c12 cells, not promotion of quiescence, Reference #22 does not support the claim that hypoxia induces myostatin expression, only that myostatin inhibits MyoD expression.

→ We apologize for the lack of clarity in some references that led to confused interpretations. When possible, we have replaced review articles by primary sources as requested by the reviewer.

Concerning Reference #13 (PMID: 19386789) in the initial manuscript, Yang and al. shows in Fig 2 that after muscle injury, HIF-1 α protein level is higher at 2 and 3dpi than at 0 dpi, demonstrating HIF-1 α is stabilized in injured TAs post-injury.

Regarding Reference #15 (PMID: 16256737) in the initial manuscript, the authors showed that hypoxia maintains C2C12 cells as well as primary satellite cells in an undifferentiated state (Fig 1) through HIF-dependent upregulation of Hey-2 and Hes1, which are two key targets of the Notch signaling implicated in satellite cell quiescence (Baghdadi et al., 2018 ; PMID: 29795344). Reference #15 (PMID: 12244043) in the initial manuscript, thus argues for a role of hypoxia in promoting in vitro the quiescence of MuSCs through HIF-1 α stabilization and Notch signaling. We have decided to maintain this reference in our manuscript.

Regarding Reference #22 in the initial manuscript, we agree that this article only show that myostatin inhibits MyoD expression. On the other hand, myostatin has been shown by Hayot et al., 2011 (PMID : 20884321) to be upregulated in skeletal muscle in response to hypoxic stimuli, suggesting a link between myostatin upregulation, hypoxia and MyoD inhibition. To avoid any confusion, we have decided to exclude this reference.

4) Figure 1E and 1F, does the dye intensity change with it being more accessible to the muscle during early injury as opposed to later recovery. Also, when using the probe for hypoxia determination, the whole tissue is fluorescing intensely suggesting potential non specificity. It would be prudent to use markers of hypoxia on western blots or gene expression to corroborate this data.

→ In the literature, pimonidazole is a well-known probe that has been used *in vivo* for measuring intratissular hypoxia in several experimental settings, including in injured skeletal muscles or post-ischemic hearts (Drouin et al., 2019 PMID: 31217019 ; Zampino et al., 2006 PMID: 16603692). To address the kinetics of pimonidazole staining, we have performed earlier time points of the hypoxic probe staining and demonstrated that hypoxia in the injured muscle is transient, peaking at 5 dpi with a progressive reoxygenation from 5 to 28 dpi. As the pimonidazole staining is lower at 3 dpi than 5 dpi, it is not in favor of an increased pimonidazole accessibility in early injury. Regarding the reviewer request on the potential non specificity of the probe, we are confident with our results described in Figure 1E and 1F. Indeed, immunofluorescent control staining using either the secondary antibody alone or without antibody (no primary and no secondary antibodies) on 5dpi injured TA cross-sections revealed no unspecific red signal (Figure #3 below), confirming the specificity of the hypoxia probe. Furthermore, the kinetics of muscle hypoxic environment during the regenerative process correlate with the kinetics of expression of known hypoxia target genes (Loxl-2, Pdgfb, Angpt2) (as shown in Figure 1I).

Figure #3 : Representative pictures of Pimonidazole (red) and DAPI (blue) staining on CTX-injured TA cross-sections at 5 days post-injury (dpi).

5) a) It is well known that CTX injury does not cause damage to the vasculature but directly to the muscle (Tatsumi et al doi:10.1002/stem.2639; Ramadasan-Nair et al doi:10.1074/jbc.M113.493270; Ohtsubo et al. doi: 10.1016/j.biocel.2017.02.005; Wang et al doi: 10.3390/ijms232113380). How do the authors reconcile their findings that there is vasculature damage with CTX (Fig. 1C).

b) Moreover, the endothelial cell staining (Fig. 1B) appears to be unchanged in the time course of injury. To prove vascular damage this data should be corroborated, for example with lectin perfusion.

→ a) We respectfully disagree with the reviewer's assertion that CTX injury does not cause damage to the vasculature, and none of the above references cited by the reviewer address this issue. In contrast, many other studies, in agreement with our findings, have shown that the vasculature is rapidly disrupted after

CTX-, BaCl₂- or crush-mediated muscle injury and is fully restored approximately 21 days post-injury (Hardy et al., 2016 ; PMID: 26807982; Latroche et al., 2017 ; PMID: 29198825). We are thus confident that CTX directly severely impact muscle vasculature.

b) Regarding the endothelial cell staining, our data clearly show a rapid decrease in muscle capillary density at 3dpi (Figure 1B, C), that gradually returns to normal from 5dpi to 14dpi. The rarefaction of the vascular network after injury was confirmed in 3D images obtained from CD31-stained cleared muscles (Supplemental Figure 1B). As suggested by the reviewer, in the revised manuscript we also analyzed the vascular perfusion using lectin dye at 0, 3, 5 and 14dpi (Supplemental Figure 1B, results section page 5, lines 7-9 ; Methods section page 37, lines 8-31). Our data showed that disruption of the vascular network was accompanied by a strong decrease in muscle vascular perfusion at 3dpi, whereas muscle reperfusion was restored concomitantly with muscle revascularization (Supplemental Figure 1B). Therefore, CTX injury clearly affects both muscle fibers and the vasculature, leading in decreased perfusion and low oxygen tension in the injured muscle.

6) Problems with Figure 3J. There are data points with zero clusters/isolated myofibres suggesting that the hypoxic environment caused MuSCs to not activate from quiescence. There are several outliers for example at 1% there is a zero reading that makes the data significant.

→ It is true that in Fig.3J, no cluster was evidenced per myofibers in one sample exposed to hypoxia. As stated in the Material and Methods section, a cluster is defined as an accumulation of more than 2 myogenic cells. In this sample, myogenic cells were organized in doublets and not in clusters along the fiber, which explains why we had zero clusters/isolated myofiber. Nevertheless, the presence of doublets clearly indicates the exit of satellite cells from quiescence towards activation. Hence, even if we remove this outlier point, the interpretation and significance of our results remains the same, demonstrating the strength of our findings (Figure #4).

Figure #4: Quantification of the number of doublets and clusters per myofiber and the number of cells per cluster on isolated myofibers after 72h of culture under 8%, 1% or 1-8%O₂. Data are means ± SEM.

7) In Figure 1G, Loxl2 after 14 days appears to be significant, as the error bars at 0 and 14 days do not overlap and thus it does not return to normal. An n=3 is not sufficient, as one of the data points at 14 days appears to be an outlier (the data stretching from 1500 to 3000).

→ Regardless of the genes examined, we found no statistical differences between 0 and 14dpi. The large error bars reflect the heterogeneity of muscle repair in our experimental settings. As this is not a key focus of our study, we do not plan to increase the number of samples analyzed in this experiment in order to comply with the 3Rs ethical principles.

8) In Fig. 2C and 2D, there are no control CSA and myofiber diameter experiments for keeping the mice in hypoxia over 14 and 28 days without injury.

→ To address the reviewer's concern, we have quantified the effect of prolonged hypoxia on the myofiber area of uninjured control TAs. Our results showed no difference in myofiber area in mice exposed to chronic hypoxia, demonstrating that prolonged hypoxia specifically affects the myofiber size only in regenerating TAs. This new analysis has been added to the text as Supplemental Figure 3B and results section page 6, lines 17-21.

9) For Figure 3K, how can self-renewing MuSCs be distinguished from MuSCs that never activated? Especially in the 1% O₂ condition where few clusters formed. How does hypoxia influence activation? A 4hr or 8hr timepoint is necessary, as well as 24hrs. Also, for Figure 5E and 5F, it is possible that HIFcKO allowed the cells to activate normally, thus explaining the shift from quiescence to activation in the read-outs. This further highlights the importance of analyzing earlier timepoints. One cannot state that these cells are self-renewing or returning to quiescence without performing experiments on earlier timepoints.

→ We thank the reviewer for raising this interesting point. In the literature, Zammit et al, 2004 (doi: 10.1083/jcb.200312007) have shown that after 48h of isolated fiber culture in normoxia (20% O₂), 98% of the satellite cell pool is activated, co-expressing Pax7 and MyoD, and that after 72h of culture, the number of Pax7⁺/MyoD⁻ cells reflected the proportion of activated myogenic cells that had returned to quiescence to maintain the satellite cell pool. To demonstrate that it also the case under hypoxic conditions, we analyzed the percentage of quiescent (PAX7⁺MYOD⁻), activated (PAX7⁺MYOD⁺) and differentiated myogenic cells after 48h of culture. We showed that after 48h of culture under hypoxia (1%O₂), only 2% of cells remained in their quiescent state whereas most of the myogenic cells along the isolated fibers underwent activation or differentiation. This supports the idea that the hypoxic condition does not affect the ability of satellite cells to fully activate ex vivo and that the quantification of PAX7⁺/MYOD⁻ cells at 72h undoubtedly reflects the ability of myogenic cells to return to a quiescent state (Figure #5).

Figure #5: Quantification of the percentage of quiescent PAX7⁺/MYOD⁻, activated PAX7⁺/MYOD⁺ and differentiated PAX7⁻/MYOD⁺ cells on isolated myofibers cultivated under 1% for 48h.

Stats: Results are expressed as means ± SEM. One-way ANOVA followed by Tukey's post-test. n=5 per O₂ level.

10) The data for Figure 4 does not suggest that transient reoxygenation is required “for proper skeletal muscle repair” as stated by the authors only that reoxygenation has rescued the phenotype in the primary myoblasts. There is no hypoxia in the control (8% O₂) for regeneration to occur (Fig. 2B).

→ We agree with the reviewer. We have modified the text accordingly (results section page 9, lines 4-6): “In contrast, transient hypoxia followed by progressive reoxygenation rescued the fusion capacity of MuSCs (Fig. 4B and 4C)”.

11) One cannot rule out metabolic dysregulation. It’s true that glycolytic fibers are generally larger than oxidative, it is likely that that alone does not explain the difference in fiber size. However, the fact that the fibers are more glycolytic does suggest a metabolic shift in the muscle (which was the aim of the experiment), which could also shift MuSC character altering their behaviour. How are MuSCs metabolically responding to hypoxia?

→ We agree with the reviewer that the results presented in the initial manuscript lacked sufficient detail to definitively rule out metabolic dysregulation. To address this, we sought to further characterize the metabolic differences between regenerated fibers in normoxic and hypoxic environments. Previous reports have indicated that in certain cases, a change in the myosin profile of a fiber is not necessarily linked to alterations in its metabolic properties (Castets et al., 2013 PMID: 23602450 ; Prola et al., 2021 PMID: 33523852). To investigate this, we performed succinate dehydrogenase (SDH) enzymatic activity staining, a well-established marker of mitochondrial oxidative capacity. In this assay, oxidative fibers are identified by their dark staining, indicative of high SDH activity, while glycolytic fibers exhibit lighter staining. As shown in supplemental figure 3F, no visible differences in SDH staining were observed between regenerated fibers at 7 dpi in mice exposed to either normoxia or prolonged hypoxia. This result has been included in the manuscript as follows (results section, page 6 line 34 and page 7 lines 1-9) :

“Fiber type analysis showed a lower percentage of type-IIA fibers and a higher percentage of type-IIB fibers in hypoxic-exposed regenerating muscles at 28 dpi (Fig. 2G and 2H). These data demonstrate that sustained hypoxia induces ~~an oxidative to glycolytic~~ a transition from slow to fast new-formed myofibers after an injury. However, staining for succinate dehydrogenase (SDH) activity did not reveal any metabolic alterations under these conditions (Supplemental figure 3F). Since glycolytic fast fibers are larger than slow oxidative fibers, it is unlikely that the glycolytic myosin switch observed under hypoxic conditions is responsible for reducing muscle mass and myofiber size.”

Next, to evaluate whether hypoxia exposure induces metabolic changes in MuSCs, we conducted a non-targeted large-scale metabolomic analysis on MuSCs isolated from regenerated muscles at 7dpi in mice exposed to either normoxia or prolonged hypoxia. Among the 84 identified metabolites, none were significantly altered by the hypoxic environment (Supplemental Figure 3G). This finding rules out the possibility that a shift in MuSC metabolism underlies the observed phenotype and changes in their behavior. These new results, which improve the understanding of the mechanisms regulating the behavior of satellite cells under prolonged hypoxia, have been added to the revised version of the article (Supplemental Figure 3G ; result section page 9, lines 17-23 ; methods section page 38, lines 1-15).

12) In Figure 2, how can one be sure that reoxygenation is blocked by the hypoxic chamber? Reduced O₂ levels will induce hypoxia, but one cannot state that it blocks reoxygenation without further

validation such as using pimonidazole as in Fig. 1E. If reoxygenation is blocked, then pimonidazole staining should remain consistent throughout the injury.

→ This point has already been raised by another reviewer. We have added complementary data on representative pimonidazole staining and quantification in mice exposed to hypoxia (Updated Supplemental Figure 2C, D). Our data show that when mice were housed in hypoxia, the intensity of the hypoxic probe staining remained greater at all time points in regenerating TAs (Updated Supplemental Figure 2C-D), demonstrating that muscle reoxygenation during regeneration is hampered when the mice are housed in hypoxic chambers. These new data have been incorporated in the revised manuscript (Supplemental Figure 2C-D ; results section page 6, lines 9-12).

13) For Figure 3G, is a sum appropriate for the graph? Proportions would be more appropriate as cell number is not equal as shown in figure 3E. Can Pax7+/MyoD+ be defined as differentiated? By day 7, many MuSCs will have fused and be expressing MyoG, which is not accounted for by these definitions. Did systemic hypoxia increase self-renewal or impair activation? How can you distinguish these two?

→ We thank the reviewer for these comments. Regarding the presentation of Figure 3G, as suggested, we modified the graph to show the proportions of each cell population rather than the cell numbers.

The PAX7+/MYOD+ cells cannot be defined as solely differentiated cells since some cells could have lost the expression of MYOD and return into quiescence. Consequently, we routinely defined in our lab : PAX7+ MYOD- KI67- cells, as quiescent cells; PAX7+ MYOD+ KI67- cells, as activated cells; PAX7+ MYOD+ KI67+ cells, as proliferative cells; PAX7- MYOD+ KI67- cells, as differentiated cells

Moreover, we agree with reviewer that at 7 dpi, many myogenic cells have already begun to fuse. *In vivo*, MYOG expression in mononucleated myogenic cells is transient, as these cells rapidly fuse into newly formed myofibers once MYOG is expressed. For this reason, we avoid using MYOG as a marker of differentiated myogenic cells *in vivo*, as its brief expression window may lead to challenges in data interpretation.

Regarding the last comment of the reviewer, approximately 90% of PAX7+ cells are all activated and in proliferation in injured muscle at 3 dpi (Figure 1F), reaching a peak of accumulated PAX7+ cells at 5 dpi in both hypoxic and normoxic conditions (Figure 3A). These data argue for an increased self-renewal of PAX7+ cells at 7 dpi under systemic hypoxia rather than an impaired activation.

14) In Figure 6A, while it is interesting that Pax7 levels are elevated in hypoxia and differentiation and fusion markers are down at 7days, it does not necessarily mean that self-renewal is increased. It might suggest that the hypoxic cells might have never activated or might have differentiated precociously. Are any cell cycle genes down regulated? Any other genes involved in quiescence altered?

→ According to our RNAseq data, no cell cycle-related genes were significantly down-regulated by hypoxia ($\log_2FC > \pm 0.5$; $p < 0.05$) nor were any genes associated with quiescence altered. However, it is important to note that this analysis was performed at 7 dpi, a time point that follows the peak of PAX7+ cell activation, proliferation, and early differentiation, corresponding to a regenerative phase in which satellite cells primarily contribute to self-renewing the muscle stem cell pool. This timing likely explains the absence of detectable transcriptional changes in cell cycle or quiescence-associated genes.

15) The use of pimonidazole in Fig. 1E shows the staining within fibers (many with centrally located nuclei). These nuclei are differentiating and not representative of expanding MuSCs. How do the authors reconcile these MuSCs as part of their population.

→ We do agree with the reviewer that Figure 1E do not show that expanding MuSCs are in hypoxic environment. However, Figure 3B does. Indeed, by performing co-immunostaining of PAX7 and pimonidazole, we were able to demonstrate the PAX7+ cells at 5 dpi are co-stained with pimonidazole adduct and thus located in a hypoxic environment.

Minor Problems:

1) In the introduction, the line "Vascular alterations result in reduced oxygen (O₂) levels, disrupting cell homeostasis and contributing to many diseases" is not always true as vascular alterations do not always result in reduced oxygen levels. For example, in angiogenesis there is no reduction of O₂. This line should better reflect this.

→ We do agree that physiological angiogenesis is not associated with reduction of O₂, as opposed to pathological angiogenesis which is frequently linked to tissue hypoxia. To clarify this point, we have revised the corresponding sentence in the introduction: "Vascular alterations is often associated with reduced oxygen (O₂) levels, disrupting cell homeostasis and contributing to many diseases" (Introduction section, page 3, lines 15-17).

2) In the introduction, Paragraph 2, line 9 change "quiescence thought HIF-1 α " to "quiescence through HIF-1 α ".

→ The change has been made (Introduction section, page 3, line 21).

3) Paragraph 3, line 8: "lead" instead of "leads"

→ The change has been made (Introduction section, page 3, line 33).

4) It is not sure how important the connection between capillary density and Pax7+ cell number is. Both are presumed to occur at the same time in muscle, so both will recover concurrently. To state that it is a coupled response is overstating the evidence presented.

→ In the muscle, CTX injection disrupts both the vasculature and the myofibers, leading to a concomitant myogenesis and angiogenesis coupling for proper muscle repair, as already described by other lab in the literature (Latroche et al, 2017, PMID: 29198825 ; Koike et al., 2022 ; PMID: 36380396). In contrast, in the mouse hindlimb ischemic model that affects primarily the muscle vasculature and secondly the myofibers, angiogenesis takes place earlier than myogenesis to orchestrate muscle repair (data not shown). This result argues for a coupled response of endothelial and satellite cells in triggering proper muscle repair after CTX-induced muscle damage.

5) Figure 1B the colour-labels for Pax7 and Dapi over lap with the border.

→ This legend of Figure 2B has been corrected in the updated version of the manuscript.

6) In the Introduction, the following sentence does not follow the previous sentence: "In vivo, Majmundar and colleagues show that HIF-1 α in MuSCs negatively regulates myogenesis by decreasing myogenic differentiation".

→ We apologize for the misunderstanding. This section of the introduction has been revised for clarity (Introduction section, page 3, lines 17-26).

7) In the Introduction, the following statement is not accurate "Hypoxia can also alter myogenic differentiation and myotube formation by inhibiting p21 (as known as p21 and CDKN1A)", for what was found in the reference. The authors should correct this statement.

→ Thus, hypoxia, by inhibiting MyoD expression, blocked accumulation of early myogenic differentiation markers such as myogenin and p21 and prevented both cell cycle withdraw and terminal differentiation. This statement was corrected in the updated version of the manuscript by "Hypoxia can also alter myogenic differentiation and myotube formation by inhibiting p21 (also known as CDKN1A) independently of HIF pathway" (Introduction section, page 3, line 31).

8) Paragraph 3, line 5: "as known as p21 and CDKN1A" should perhaps read "also known as CDKN1A"

→ The text has been revised in the updated version of the manuscript (Introduction section, page 3, line 31).

9) The following statement is not supported by the results: "Strikingly, the most abundant and intense pimonidazole staining is detected on CTX-injured TAs at 5 dpi, indicating that myogenic cell expansion is initiated in a hypoxic environment *in situ* (Fig. 1D-1F)." MuSCs are activated and expanding from time zero to 5 days according to Figure 1D.

→ To support our statement, we have assessed the kinetics of satellite cells proliferation during muscle regeneration (Figure 1F, results section page 5, lines 13-16). Our results show that myogenic cell proliferation is maximal at 3dpi, with more than 80% of the PAX7⁺ cells being also KI67⁺. This percentage decreases to 52% at 5dpi, whereas muscle hypoxia increased at 3dpi to reach a plateau at 5dpi (Figure 1D-H). These data support our statement that myogenic cell expansion is initiated in low oxygen environment *in situ*.

10) "...Since glycolytic fibers are larger than oxidative fibers, ..." citation missing

→ We apologize for the missing citation that has been updated in the revised manuscript (Liu et al., 2013 PMID: 24092696 ; References section page 44, lines 3-4).

11) An inconsistent finding is that the authors show that protein synthesis rates are normal between normoxia and hypoxia of regenerating muscle (suppl. Fig. 1E), yet the capacity of protein synthesis is found to be higher in oxidative muscle fibres compared to glycolytic fibers (Van Wessel et al, doi: 10.1007/s00421-010-1545-0), which are formed during regeneration (Fig. 2G and 2H).

→ One explanation for these inconsistent findings is that, although systemic hypoxia leads to a transition from slow to fast newly formed myofibers after injury, there is no evidence of metabolic alterations in myofibers exposed to hypoxia (Supplemental Figure 3F). This lack of a metabolic switch in the myofibers

could explain why protein synthesis rates appear similar in regenerating muscle under normoxic and hypoxic exposure. These new results and interpretation have been added in the revised manuscript (Supplemental Figure 3F., results section page 7, lines 4-7).

12) Some figure legends that describe graphs do not denote the number of samples or mice used.

→ The number of samples is indicated in the “Statistics” part of every Figure legend.

13) In Figure 1C, 1D and 1F what is being compared to obtain statistical significance?

→ As described in the “Statistics” part of the legend, we compared each time point post-dpi to the one at 0dpi.

14) The font size of many figures is too small to follow.

→ We apologize for this font size issue. Font size of figures has been increased in the revised manuscript.

15) Confusion for the results of figure 3G. Labels in the text do not reflect the labels in figure (which cannot be read anyway because the font is too small). Why is Ki67 used as a marker for activation versus proliferation.

→ Once again, we apologize for the small font size. Ki67 is used as a marker of proliferation whereas MYOD is used as a marker of activation. We thus consider that PAX7⁺ MYOD⁺ Ki67⁻ cells represent the activated myogenic cells whereas PAX7⁺ MYOD⁺ Ki67⁺ cells reflect the proliferating myogenic cells.

16) The physiological O₂ concentration is 8%, do the authors know what the hypoxic O₂ concentration is in the injured environment. Why did they choose hypoxic O₂ concentration at 1% for ex vivo and invitro experiments? Why did they choose 10% for the in vivo experiment?

→ We do not know the precise oxygen concentration in the injured TA muscles in situ. However, we do know that the detection threshold of the hypoxia probe is 1% or less. For our ex vivo and in vitro experiments, we therefore selected a concentration of 1% to mimic that found in injured muscles. Salyhaa and Oliynyk (PMID: 36718422) indicate that, in unadapted animals, signs of hypoxia begin to develop below 14% oxygen in the atmosphere. We therefore chose to expose our mice to 10% atmospheric O₂, which is well tolerated and sufficient to induce systemic hypoxia and slow muscle reoxygenation after an injury (Supplemental Figure 2).

17) For Figure 2H it is not appropriate to state that type IIA ratio was reduced with hypoxia, as the results show no statistical significance.

→ We do agree with the reviewer that Figure 2H only show a trend in decreasing the % of type-IIA under hypoxia but as in Supplemental figure 5H this trend appears significant in HIF CTRL mice exposed to 10%O₂ we decided to keep our statement in the revised manuscript.

18) For Figure legend 3K, are the cell number/fiber the sums per one mouse or the sum from all mice combined for each condition?

→ For figure legend 3K, the cell number/myofiber is the sum from all 4 mice for each condition.

19) For Figure 3B and 3E "concomitantly with their proliferation peak" seems to imply that hypoxia in Pax7+ cells peaks alongside proliferation, but the evidence doesn't support that conclusion. More timepoints would be needed to show that 5 dpi is truly the peak of hypoxia in Pax7+ cells.

→ At the reviewer's request, we analyzed an earlier time point than 5 dpi in order to further define the peak of hypoxia in PAX7+ cells. As shown in Figures 1E and 1F, although the percentage of proliferating PAX7+KI67+ cells peaks at 3 dpi, the accumulation of PAX7+ cells in injured TAs is maximal at 5dpi (Figure 1F), coinciding with the maximum intensity of hypoxia probe staining (Figure 1H). These new data support our hypothesis that 5 dpi indeed corresponds to the peak of hypoxia in PAX7+ cells. In the revised manuscript, the sentence has been corrected by "concomitantly with their expansion peak" (results section page 7, lines 26-27).

20) For Figure legend 4E, should read "MHC" not "MCH"

→ We apologize for this mistake, that has been corrected in the updated version of the manuscript.

21) In Figure 4C there is no gap between the significance bar.

→ This format error has been corrected in the updated figure.

22) In Figure legend 5G, "Experience design" should read "Experimental design"

→ The text has been corrected in the updated version of the manuscript.

23) Representative images Fig 3I and 5E are poor quality.

→ We apologize for the poor quality of our representative images, which have been taken with Zeiss AxioImager D1 microscope. Unfortunately, these images cannot be replaced.

24) Confusing statement "In the same way, this presence of smaller myofibers under prolonged hypoxia could not be explain by the glycolytic fiber-type switch from type-IIA to type-IIB, as observed in pathological context of COPD or peripheral arterial disease (PAD), since type-IIB are the largest myofibers in mice."

We do agree with the reviewer that comparing our results with the pathological context of COPD or PAD is confusing. We have corrected this in the new manuscript (discussion section, page 15, lines 5-8).

Dear Dr. GERVAIS,

Thank you for your patience while your revised manuscript was re-reviewed. We have now received the enclosed reports from the referees and I am happy to say that all support its publication now. Referees 2 and 3 still have a few more minor suggestions that I would like you to incorporate before we can proceed with the official acceptance of your manuscript.

A few editorial requests will also need to be addressed:

- Please submit the final ms file as a Word file. The figures need to be uploaded as individual production quality Figure files and we only need figure legends in the ms, after the References.
- Please add a Data Availability Section (DAS) to the end of the Methods. The DAS should list links to data generated in this study and deposited in public databases. If no such data were generated, this needs to be mentioned in the DAS.
- Please correct the conflict of interest subheading to "Disclosure and Competing Interests Statement"
- The author affiliations all need to be provided on the title page below the author list, not as footnotes.
- The author credits need to be removed from the ms file. All credits need to be entered during online ms submission.
- The REFERENCE format needs to be alphabetical, not numerical; et al needs to be used after 10 author names. Please use the EMBO reports reference style.
- Please send us with your final ms a completed author checklist, which you can download from our author guidelines <<https://www.embopress.org/page/journal/14693178/authorguide>>. The completed author checklist will also be part of the transparent peer-review file.
- The FUNDING INFO is missing in our online submission system, please add also all funding info there. The funders acknowledged in the ms need to be entered separately under Acknowledgements (no comments box should be used).
- A callout for Figure 4G is missing, please add.
- Instead of a supplemental file, you can upload the supplemental figures as individual Expanded View (EV) figures. Please have a look at our guide to authors online for more information. The EV figures are integrated inline in the html version of your paper and thus slightly more accessible/visible. The EV figures legends should be listed after the main figure legends in the main ms file. All figure callouts need to be amended.

If you prefer to keep a separate file for these data, it should be called APPENDIX FILE and uploaded as a single pdf file. "Supplemental" should not be used at all; the title page of the Appendix should have a table of contents with page numbers; the figures and tables should be updated to Appendix Figure S1, etc. Appendix Table S1, etc. and the ms callouts need to be corrected too and every occurrence of "supplemental" should be removed/updated; each figure legend should follow its figure.

- The supplemental tables should be part of the Reagents & Tools table.

All Materials and Methods need to be described in the main text and the Methods section should include a separate Reagents and Tools Table file (listing key reagents, experimental models, software and relevant equipment and including their sources and relevant identifiers) and a Methods and Protocols section in which we ask authors to describe their methods using a step-by-step protocol format with bullet points, to facilitate the adoption of the methodologies across labs. More information on how to adhere to this format as well as downloadable templates (.docx) for the Reagents and Tools Table can be found in our author guidelines: < <https://www.embopress.org/page/journal/14693178/authorguide#manuscriptpreparation>>.

- Summary should be Abstract
- Correspondence section should be removed from the ms
- The manuscript sections should be in the following order: Title page - Abstract & Keywords - Introduction - Results - Discussion - Methods - Data Availability - Acknowledgments - Disclosure Statement & Competing Interests - References - Figure Legends - (Main Tables with legends if applicable) - Expanded View Figure Legends.

Figure Legends - Comments

- Please note that the figure 7F, G is mislabeled as figure 7K, L in the manuscript. This needs to be rectified.
- Please note that the exact p values are not provided in the legends of figures 1C, D, F, H; 4C, 7J, please provide exact values as reasonable.
- Please indicate the statistical test used for data analysis in the legends of figures 1C, 4D
- Please note that information related to n is missing in the legends of figures 1C, 4D
- Please note that the error bars are not defined in the legends of figures 1C, 4D

EMBO press papers are accompanied online by A) a short (1-2 sentences) summary of the findings and their significance, B) 2-3 bullet points highlighting key results and C) a synopsis image that is exactly 550 pixels wide and 200-600 pixels high (the height is variable). The synopsis image should provide a sketch of the major findings, like a graphical abstract. Please note that text needs to be readable at the final size. Please send us this information along with the final manuscript.

Referee #1:

The authors have met most of my concerns.

Referee #2:

The authors have done a good job in addressing my concerns.
I have only two remaining aspects.

1. In Figure 1B, I assume that the broad cytoplasmic staining seen with the mouse anti-Pax7 antibody is a non-specific antibody uptake due to the necrosis of the fibers? If so, the authors could repeat the staining using an anti-Pax7 antibody not made in mouse or at least the reason for the odd signal should be clarified in the text.
2. In Figure 2B, I don't see any difference between the old and the new version of the images for the hypoxia condition.

Referee #3:

The authors have answered my previous critiques of the manuscript and I think this paper will be an important contribution. There is only one change that was not made: Fig 2C still shows myofiber diameter and not cross-sectional area. Please replace with the correct graph.

To answer to the referee :

Referee #1:

The authors have met most of my concerns.

Referee #2:

The authors have done a good job in addressing my concerns.

I have only two remaining aspects.

1. In Figure 1B, I assume that the broad cytoplasmic staining seen with the mouse anti-Pax7 antibody is a non-specific antibody uptake due to the necrosis of the fibers? If so, the authors could repeat the staining using an anti-Pax7 antibody not made in mouse or at least the reason for the odd signal should be clarified in the text.

2. In Figure 2B, I don't see any difference between the old and the new version of the images for the hypoxia condition.

Referee #3:

The authors have answered my previous critiques of the manuscript and I think this paper will be an important contribution. There is only one change that was not made: Fig 2C still shows myofiber diameter and not cross-sectional area. Please replace with the correct graph.

→ We would like to thank all the referees for their feedback.

As suggested by referee #2, the non-specific myofiber staining observed in Figure 1B is due to an unspecific uptake of the anti-mouse PAX7 antibody that marks IgG+ necrotic fibers. This has been clarified in the Figure 1B legend.

We apologize for this mistake. The correct Figure 2 has been added in the former ms. This new version addresses the concerns raised by referees #2 and #3.

Regarding some of your editorial requests, please our answer in blue:

- Correspondence section should be removed from the ms

→ This section has been removed from the ms.

- The manuscript sections should be in the following order: Title page - Abstract & Keywords - Introduction - Results - Discussion - Methods - Data Availability - Acknowledgments - Disclosure Statement & Competing Interests - References - Figure Legends - (Main Tables with legends if applicable) - Expanded View Figure Legends.

→ The ms has been reorganized accordingly to the editorial request.

- A callout for Figure 4G is missing, please add.

→ We apologize for this error. The callout for Figure 4G has now incorporated in the revised ms.

- Summary should be Abstract.

→ As requested, this section has been renamed.

- Correspondence section should be removed from the ms.

→ The correspondence section has been removed from the ms.

Figure Legends - Comments

- Please note that the figure 7F, G is mislabeled as figure 7K, L in the manuscript. This needs to be rectified.

→ This mislabeling has been corrected in the revised manuscript.

- Please note that the exact p values are not provided in the legends of figures 1C, D, F, H; 4C, 7J, please provide exact values as reasonable.

→ Using Graphpad software, we do not have access to the exact p value when the p value is under $p < 0.0001$.

- Please indicate the statistical test used for data analysis in the legends of figures 1C, 4D

- Please note that information related to n is missing in the legends of figures 1C, 4D

- Please note that the error bars are not defined in the legends of figures 1C, 4D

→ For Figures 1C and 4D, the figure legends have been modified, accordingly to your requests.

Regarding the routine image analysis:

1. Possible reuse of cells between Figure 4B, first cell, and Figure 5H first cell, top lane. Can you please clarify?

→ We apologize for this mistake. The image top panel of figure 5H has been replaced and modified accordingly.

2. Appendix Figure 5G - Top / 1st cell has a repeating background. Please send us source data for this image and explain/clarify.

→ The repeated background in Figure 5G top panel is linked to the use of “tile mode” of Zeiss AxioImager D1 microscope to image sampling and mainly due to a poor shading correction at the edge of individual tiles. This background noise effect was even amplified by the low-quality of image export. Please file attach the original image (HIF643_Fiber types x10.czi) that you can open with QuPath software.

Dr. Marianne GERVAIS
Mondor Institute of Biomedical Research
INSERM U955
France

Dear Dr. GERVAIS,

I am very pleased to accept your manuscript for publication in the next available issue of EMBO reports. Thank you for your contribution to our journal.

You may qualify for financial assistance for your publication charges - either via a Springer Nature fully open access agreement or an EMBO initiative. Check your eligibility: <https://link.springer.com/journal/44319/how-to-publish-with-us>

Yours sincerely,

>>> Please note that it is EMBO Reports policy for the transcript of the editorial process (containing referee reports and your response letter) to be published as an online supplement to each paper. If you do NOT want this, you will need to inform the Editorial Office via email immediately. More information is available here: <https://link.springer.com/partners/embo-press/editorial-policies#Peer%20review>